# Clearer Frames, Anytime: Resolving Velocity Ambiguity in Video Frame Interpolation

## Abstract

Existing video frame interpolation (VFI) methods blindly predict where each object is at a specific timestep $t$ ("time indexing"), which struggles to predict precise object movements. Given two images of a baseball, there are infinitely many possible trajectories: accelerating or decelerating, straight or curved. This often results in blurry frames as the method averages out these possibilities. Instead of forcing the network to learn this complicated time-to-location mapping implicitly together with predicting the frames, we provide the network with an explicit hint on how far the object has traveled between start and end frames, a novel approach termed "distance indexing". This method offers a clearer learning goal for models, reducing the uncertainty tied to object speeds. We further observed that, even with this extra guidance, objects can still be blurry especially when they are equally far from both input frames (*i.e.*, halfway in-between), due to the directional ambiguity in long-range motion. To solve this, we propose an iterative reference-based estimation strategy that breaks down a long-range prediction into several short-range steps. When integrating our plug-and-play strategies into state-of-the-art learning-based models, they exhibit markedly sharper outputs and superior perceptual quality in arbitrary time interpolations, using a uniform distance indexing map in the same format as time indexing. Additionally, distance indexing can be specified pixel-wise, which enables temporal manipulation of each object independently, offering a novel tool for video editing tasks like re-timing.

## 1 Introduction

Video frame interpolation (VFI) plays a crucial role in creating slow-motion videos (Bao et al., 2019), video generation (Ho et al., 2022), prediction (Wu et al., 2022), and compression (Wu et al., 2018). Directly warping the starting and ending frames using the optical flow between them can only model linear motion, which often diverges from actual motion paths, leading to artifacts such as holes. To solve this, learning-based methods have emerged as leading solutions to VFI. which aim to develop a model, represented as $\mathcal{F}$, that uses a starting frame $I_0$ and an ending frame $I_1$ to generate a frame for a given timestep, described by:

$$I_t = \mathcal{F}\left(I_0, I_1, t\right). \tag{1}$$

Two paradigms have been proposed: In fixed-time interpolation (Liu et al., 2017; Bao et al., 2019), the model only takes the two frames as input and always tries to predict the frame at $t = 0.5$. In arbitrary-time interpolation (Jiang et al., 2018; Huang et al., 2022), the model is further given a user-specified timestep $t \in [0, 1]$, which is more flexible at predicting multiple frames in-between.

Yet, in both cases, the unsampled blank between the two frames, such as the motion between a ball's starting and ending points, presents infinite possibilities. The velocities of individual objects within these frames remain undefined, introducing a *velocity ambiguity*, a myriad of plausible time-to-location mappings during training. We observed that velocity ambiguity is a primary obstacle hindering the advancement of learning-based VFI: Models trained using aforementioned *time indexing* receive identical inputs with differing supervision signals during training. As a result, they tend to produce blurred and imprecise interpolations, as they average out the potential outcomes.

Could an alternative indexing method minimize such conflicts? One straightforward option is to provide the optical flow at the target timestep as an explicit hint on object motion. However, this

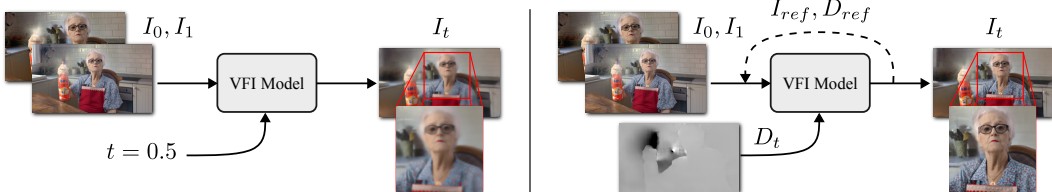

(a) Training paradigm of time indexing      (b) Training paradigm of distance indexing

Figure 1: Comparison of time indexing and distance indexing training paradigms. (a) Time indexing uses the starting frame $I_0$, ending frame $I_1$, and a scalar variable $t$ as inputs. (b) Distance indexing replaces the scalar with a distance map $D_t$ and optionally incorporates iterative reference-based estimation $(I_{ref}, D_{ref})$ to address velocity ambiguity, resulting in a notably sharper prediction.

information is unknown at inference time, which has to be approximated by the optical flow between $I_0$ and $I_1$, scaled by the timestep. This requires running optical flow estimation on top of VFI, which may increase the computational complexity and enforce the VFI algorithm to rely on the explicitly computed but approximate flow. Instead, we propose a more flexible *distance indexing* approach. In lieu of an optical flow map, we employ a *distance ratio* map $D_t$, where each pixel denotes *how far the object has traveled between start and end frames*, within a normalized range of $[0, 1]$,

$$I_t = \mathcal{F}\left(I_0, I_1, \text{explicit motion hint}\right) \quad \Rightarrow \quad I_t = \mathcal{F}\left(I_0, I_1, D_t\right). \tag{2}$$

**During training, $D_t$ is derived from optical flow ratios computed from ground-truth frames. During inference, it is sufficient to provide a uniform map as input**, in the exactly same way as time indexing methods, *i.e.*, $D_t(x, y) = t$, $\forall x, y$. However, the semantics of this indexing map have shifted from an uncertain timestep map to a more deterministic motion hint. Through distance indexing, we effectively solve the one-to-many time-to-position mapping dilemma, fostering enhanced convergence and interpolation quality.

Although distance indexing addresses the scalar *speed ambiguity*, the *directional ambiguity* of motion remains a challenge. We observed that this directional uncertainty is most pronounced when situated equally far from the two input frames, *i.e.*, halfway between them. Drawing inspiration from countless computer vision algorithms that iteratively solve a difficult problem (*e.g.*, optical flow (Teed & Deng, 2020) and image generation (Rombach et al., 2022)), we introduce an iterative reference-based estimation strategy. This strategy seeks to mitigate directional ambiguity by incrementally estimating distances, beginning with nearby points and advancing to farther ones, such that the uncertainty within each step is minimized and the image quality is further improved.

Our approach addresses challenges that are not bound to specific network architectures. Indeed, it can be applied as a plug-and-play strategy that requires only modifying the input channels for each model, as demonstrated in Figure 1. We conducted extensive experiments on four existing VFI methods to validate the effectiveness of our approach, which produces frames of markedly improved perceptual quality. Moreover, instead of using a uniform map, it is also possible to use a spatially-varying 2D map as input to manipulate the motion of objects. Paired with state-of-the-art segmentation models such as Segment Anything Model (SAM) (Kirillov et al., 2023), this empowers users to freely control the interpolation of any object, *e.g.*, making certain objects backtrack in time.

In summary, our key contributions are: 1) Proposing distance indexing and iterative reference-based estimation to address the velocity ambiguity and enhance the capabilities of arbitrary time interpolation models; 2) Conducting comprehensive validation of the efficacy of our plug-and-play strategies across a range of state-of-the-art learning-based models. 3) Presenting an unprecedented manipulation method that allows for customized interpolation of any object.

## 2 RELATED WORK

### 2.1 VIDEO FRAME INTERPOLATION

**General overview.** Numerous VFI solutions rely on optical flows to predict latent frames. Typically, these methods warp input frames forward or backward using flow calculated by off-the-shelf networks like Sun et al. (2018); Dosovitskiy et al. (2015); Ilg et al. (2017); Teed & Deng (2020) or

self-contained flow estimators like Huang et al. (2022); Zhang et al. (2023); Li et al. (2023). Networks then refine the warped frame to improve visual quality. SuperSlomo (Jiang et al., 2018) uses a linear combination of bi-directional flows for intermediate flow estimation and backward warping. DAIN (Bao et al., 2019) introduces a depth-aware flow projection layer for advanced intermediate flow estimation. AdaCoF (Lee et al., 2020) estimates kernel weights and offset vectors for each target pixel, while BMBC (Park et al., 2020) and ABME (Park et al., 2021) refine optical flow estimation. Large motion interpolation is addressed by Sim et al. (2021) through a recursive multi-scale structure. VFIFormer (Lu et al., 2022) employs Transformers to model long-range pixel correlations. IFRNet (Kong et al., 2022), RIFE (Huang et al., 2022), and UPR-Net (Jin et al., 2023) employ efficient pyramid network designs for high-quality, real-time interpolation, with IFRNet and RIFE using leakage distillation losses for flow estimation. Recently, more advanced network modules and operations are proposed to push the upper limit of VFI performance, such as the transformer-based bilateral motion estimator of BiFormer (Park et al., 2023), a unifying operation of EMA-VFI (Zhang et al., 2023) to explicitly disentangle motion and appearance information, and bi-directional correlation volumes for all pairs of pixels of AMT (Li et al., 2023). On the other hand, SoftSplat (Niklaus & Liu, 2020) and M2M (Hu et al., 2022) actively explore the forward warping operation for VFI.

Other contributions to VFI come from various perspectives. For instance, Xu et al. (2019) leverage acceleration information from neighboring frames, VideoINR (Chen et al., 2022) is the first to employ an implicit neural representation, and Lee et al. (2023) explore and address discontinuity in video frame interpolation using figure-text mixing data augmentation and a discontinuity map. Flow-free approaches have also attracted interest. SepConv (Niklaus et al., 2017) integrates motion estimation and pixel synthesis, CAIN (Choi et al., 2020) employs the PixelShuffle operation with channel attention, and FLAVR (Kalluri et al., 2023) utilizes 3D space-time convolutions. Additionally, specialized interpolation methods for anime, which often exhibit minimal textures and exaggerated motion, are proposed by AnimeInterp (Siyao et al., 2021) and Chen & Zwicker (2022).

**Learning paradigms.** One major thread of VFI methods train networks on triplet of frames, always predicting the central frame. Iterative estimation is used for interpolation ratios higher than $\times 2$. This *fixed-time* method often accumulates errors and struggles with interpolating at arbitrary continuous timesteps. Hence, models like SuperSloMo Jiang et al., 2018, DAIN (Bao et al., 2019), BMBC (Park et al., 2020), EDSC (Cheng & Chen, 2021), RIFE (Huang et al., 2022), IFRNet (Kong et al., 2022), EMA-VFI (Zhang et al., 2023), and AMT (Li et al., 2023) have adopted an *arbitrary time* interpolation paradigm. While theoretically superior, the arbitrary approach faces challenges of more complicated time-to-position mappings due to the velocity ambiguity, resulting in blurred results. This study addresses velocity ambiguity in arbitrary time interpolation and offers solutions.

Prior work by Zhou et al. (2023) identified motion ambiguity and proposed a texture consistency loss to implicitly ensure interpolated content resemblance to given frames. In contrast, we explicitly address velocity ambiguity and propose solutions. These innovations not only enhance the performance of arbitrary time VFI models but also offer advanced manipulation capabilities.

## 2.2 SEGMENT ANYTHING

The emergence of Segment Anything Model (SAM) (Kirillov et al., 2023) has marked a significant advancement in the realm of zero-shot segmentation, enabling numerous downstream applications including video tracking and segmentation (Yang et al., 2023), breakthrough mask-free inpainting techniques (Yu et al., 2023), and interactive image description generation (Wang et al., 2023). By specifying the distance indexing individually for each segment, this work introduces a pioneering application to this growing collection: Manipulated Interpolation of Anything.

## 3 VELOCITY AMBIGUITY

In this section, we begin by revisiting the time indexing paradigm. We then outline the associated velocity ambiguity, which encompasses both speed and directional ambiguities.

Figure 2 (a) shows the example of a horizontally moving baseball. Given a starting frame and an ending frame, along with a time indexing variable $t = 0.5$, the goal of a VFI model is to predict a latent frame at this particular timestep, in accordance with Eq. 1.

Although the starting and ending positions of the baseball are given, its location at $t = 0.5$ remains ambiguous due to an unknown speed distribution: The ball can be accelerating or decelerating,

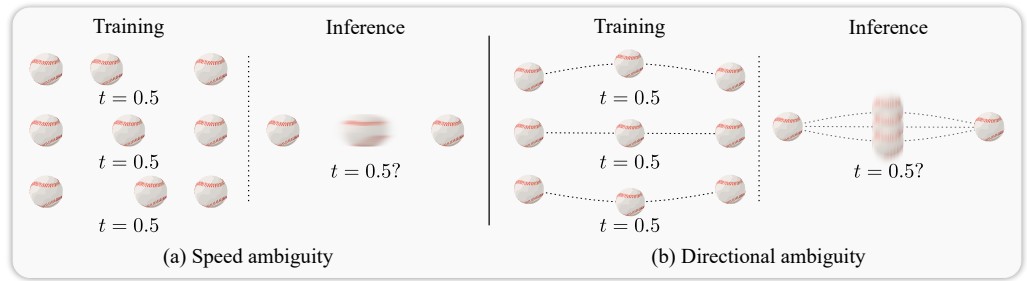

Figure 2: Velocity ambiguity. (a) Speed ambiguity. (b) Directional ambiguity.

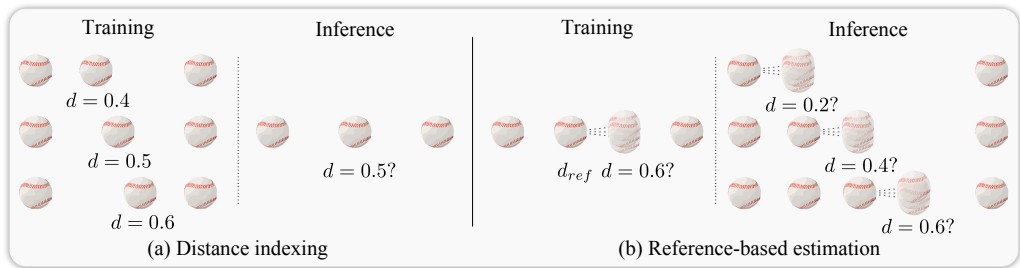

Figure 3: Disambiguation strategies for velocity ambiguity. (a) Distance indexing. (b) Iterative reference-based estimation.

resulting in different locations. This ambiguity introduces a challenge in model training as it leads to multiple valid supervision targets for the identical input. Contrary to the deterministic scenario illustrated in Eq. 1, the VFI function $\mathcal{F}$ is actually tasked with generating a *distribution* of plausible frames from the same input frames and time indexing. This can be expressed as:

$$\left\{I_t^1, I_t^2, \ldots, I_t^n\right\} = \mathcal{F}(I_0, I_1, t), \tag{3}$$

where $n$ is the number of plausible frames. Empirically, the model, when trained with this ambiguity, tends to produce a weighted average of possible frames during inference. While this minimizes the loss during training, it results in blurry frames that are perceptually unsatisfying to humans, as shown in Figure 1 (a). This blurry prediction $\hat{I}_t$ can be considered as an average over all the possibilities:

$$\hat{I}_t = \mathbb{E}_{I_t \sim \mathcal{F}(I_0, I_1, t)}[I_t]. \qquad \text{(See details in Appendix A)} \tag{4}$$

Indeed, not only the speed but also the direction of motion remains indeterminate, leading to what we term as "directional ambiguity." This phenomenon is graphically depicted in Figure 2 (b). This adds an additional layer of complexity in model training and inference. We collectively refer to speed ambiguity and directional ambiguity as velocity ambiguity.

So far, we have been discussing the ambiguity for the fixed time interpolation paradigm, in which $t$ is set by default to $0.5$. For arbitrary time interpolation, the ambiguity becomes more pronounced: Instead of predicting a single timestep, the network is expected to predict a continuum of timesteps between 0 and 1, each having a multitude of possibilities. This further complicates the learning task.

## 4  DISAMBIGUATION STRATEGIES

In this section, we introduce two innovative strategies, namely distance indexing and iterative reference-based estimation, aimed at addressing the challenges posed by the velocity ambiguity. Designed to be plug-and-play, these strategies can be seamlessly integrated into any existing VFI models without necessitating architectural modifications, as shown in Figure 1 (b).

### 4.1  DISTANCE INDEXING

In traditional time indexing, models intrinsically deduce an uncertain time-to-location mapping, represented as $\mathcal{D}$:

$$I_t = \mathcal{F}(I_0, I_1, \mathcal{D}(t)). \tag{5}$$

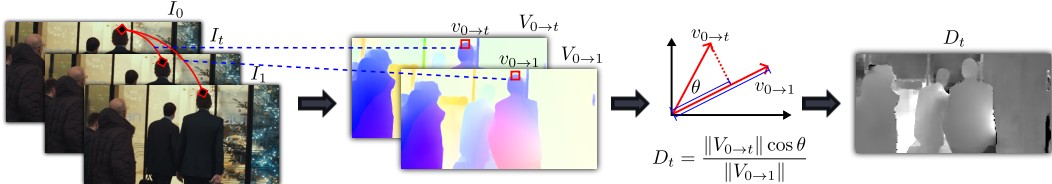

Figure 4: Calculation of distance map for distance indexing. $V_{0\to t}$ is the estimated optical flow from $I_0$ to $I_t$ by RAFT (Teed & Deng, 2020), and $V_{0\to 1}$ is the optical flow from $I_0$ to $I_1$.

This brings forth the question: Can we guide the network to interpolate more precisely without relying on the ambiguous mapping $\mathcal{D}(t)$ to decipher it independently? To address this, we introduce a strategy to diminish speed uncertainty by directly specifying a distance ratio map ($D_t$) instead of the uniform timestep map. This is termed as distance indexing. Consequently, the model sidesteps the intricate process of deducing the time-to-location mapping:

$$I_t = \mathcal{F}(I_0, I_1, D_t). \tag{6}$$

We utilize an off-the-shelf optical flow estimator, RAFT (Teed & Deng, 2020), to determine the pixel-wise distance map, as illustrated in Figure 4. Given an image triplet $\{I_0, I_1, I_t\}$, we first calculate the optical flow from $I_0$ to $I_t$, denoted as $\mathbf{V}_{0\to t}$, and from $I_0$ to $I_1$ as $\mathbf{V}_{0\to 1}$. At each pixel $(x, y)$, we project the motion vector $\mathbf{V}_{0\to t}(x, y)$ onto $\mathbf{V}_{0\to 1}(x, y)$. The distance map is then defined as the ratio between the projected $\mathbf{V}_{0\to t}(x, y)$ and $\mathbf{V}_{0\to 1}(x, y)$:

$$D_t(x, y) = \frac{\|\mathbf{V}_{0\to t}(x, y)\| \cos \theta}{\|\mathbf{V}_{0\to 1}(x, y)\|}, \tag{7}$$

where $\theta$ denotes the angle between the two. By directly integrating $D_t$, the network achieves a clear comprehension of distance during its training phase, subsequently equipping it to yield sharper frames during inference, as showcased in Figure 3 (a).

During inference, the algorithm does not have access to the exact distance map since $I_t$ is unknown. In practice, we notice it is usually sufficient to provide a uniform map $D_t = t$, similar to time indexing. Physically this encourages the model to move each object at constant speeds along their trajectories. We observe that constant speed between frames is a valid approximation for many real-world situations. In Section 5, we show that even though this results in pixel-level misalignment with the ground-truth frames, it achieves significantly higher perceptual scores and is strongly preferred in the user study. Precise distance maps can be computed from multiple frames, which can potentially further boost the performance. **Please see a detailed discussion in Appendix B.**

### 4.2 ITERATIVE REFERENCE-BASED ESTIMATION

While distance indexing addresses speed ambiguity, it omits directional information, leaving directional ambiguity unresolved. Our observations indicate that, even with distance indexing, frames predicted at greater distances from the starting and ending frames remain not clear enough due to this ambiguity. To address this, we propose an iterative reference-based estimation strategy, which divides the complex interpolation for long distances into shorter, easier steps. This strategy enhances the traditional VFI function, $\mathcal{F}$, by incorporating a reference image, $I_{\text{ref}}$, and its corresponding distance map, $D_{\text{ref}}$. Specifically, the network now takes the following channels as input:

$$I_t = \mathcal{F}(I_0, I_1, D_t, I_{\text{ref}}, D_{\text{ref}}). \tag{8}$$

For example, if we break the estimation of of a remote step $t$ into two steps:

$$I_{t/2} = \mathcal{F}(I_0, I_1, D_{t/2}, I_0, D_0). \tag{9}$$

$$I_t = \mathcal{F}(I_0, I_1, D_t, I_{t/2}, D_{t/2}). \tag{10}$$

Importantly, in every iteration, we consistently use the starting and ending frames as reliable appearance references, preventing divergence of uncertainty in each step.

By dividing a long step into shorter steps, the uncertainty in each step is reduced, as shown in Figure 3 (b). While fixed time models also employ an iterative method in a bisectioning way during inference, our strategy progresses from near to far, ensuring more deterministic trajectory interpolation. This reduces errors and uncertainties tied to a single, long-range prediction.

Our combined approach of distance indexing and iterative reference-based estimation empowers arbitrary time models to outperform fixed-time counterparts, as detailed in Appendix C.

## 5 EXPERIMENT

### 5.1 IMPLEMENTATION

We leveraged the plug-and-play nature of our distance indexing and iterative reference-based estimation strategies to seamlessly integrate them into influential arbitrary time VFI models such as RIFE (Huang et al., 2022) and IFRNet (Kong et al., 2022), and state-of-the-art models including AMT (Li et al., 2023) and EMA-VFI (Zhang et al., 2023). We adhere to the original hyperparameters for each model for a fair comparison and implement them with PyTorch (Paszke et al., 2019). For training, we use the septuplet dataset from Vimeo90K (Xue et al., 2019). The septuplet dataset comprises 91,701 seven-frame sequences at $448 \times 256$, extracted from 39,000 video clips. For evaluation, we use both pixel-centric metrics like PSNR and SSIM (Wang et al., 2004), and perceptual metrics such as reference-based LPIPS (Zhang et al., 2018) and non-reference NIQE (Mittal et al., 2012). Concerning the iterative reference-based estimation strategy, $D_{ref}$ during training is calculated from the optical flow derived from ground-truth data at a time point corresponding to a randomly selected reference frame, like $t/2$. In the inference phase, we similarly employ a uniform map for reference, for example, setting $D_{ref} = t/2$. Regarding the extra costs with our strategies, see Appendix D.

### 5.2 QUALITATIVE COMPARISON

We conducted a qualitative analysis on different variants of each arbitrary time VFI model. We evaluate the base model, labeled as $[T]$, against its enhanced versions, which incorporate distance indexing ($[D]$), iterative reference-based estimation ($[T, R]$), or a combination of both ($[D, R]$), as shown in Figure 5. We observe that the $[T]$ model yields blurry results with details difficult to distinguish. Models with the distance indexing ($[D]$) mark a noticeable enhancement in perceptual quality, presenting clearer interpolations than $[T]$. In most cases, iterative reference ($[T, R]$) also enhances model performance, with the exception of AMT-S. As expected, the combined approach $[D, R]$ offers the best quality for all base models including AMT-S. This highlights the synergistic potential of distance indexing when paired with iterative reference-based estimation. Overall, our findings underscore the effectiveness of both techniques as plug-and-play strategies, capable of significantly elevating the qualitative performance of cutting-edge arbitrary time VFI models.

### 5.3 QUANTITATIVE COMPARISON

To further substantiate the efficacy of our proposed strategies, we also conducted a quantitative analysis. Figure 6 shows the convergence curves for different model variants, *i.e.*, $[T]$, $[D]$, and $[D, R]$. The observed trends are consistent with our theoretical analysis from Section 4, supporting the premise that by addressing velocity ambiguity, both distance indexing and iterative reference-based estimation can enhance convergence limits.

In Table 1, we provide a performance breakdown for each model variant. The models $[D]$ and $[D, R]$ in the upper half utilize ground-truth distance guidance, which is not available at inference in practice. The goal here is just to show the achievable upper-bound performance. On both pixel-centric metrics such as PSNR and SSIM, and perceptual measures like LPIPS and NIQE, the improved versions $[D]$ and $[D, R]$ outperform the base model $[T]$. Notably, the combined model $[D, R]$ using both distance indexing and iterative reference-based estimation strategies performs superior in perceptual metrics, particularly NIQE. The superior pixel-centric scores of model $[D]$ compared to model $[D, R]$ can be attributed to the indirect estimation (2 iterations) in the latter, causing slight misalignment with the ground-truth, albeit with enhanced details.

In realistic scenarios where the precise distance map is inaccessible at inference, one could resort to a uniform map akin to time indexing. The bottom segment of Table 1 shows the performance of the

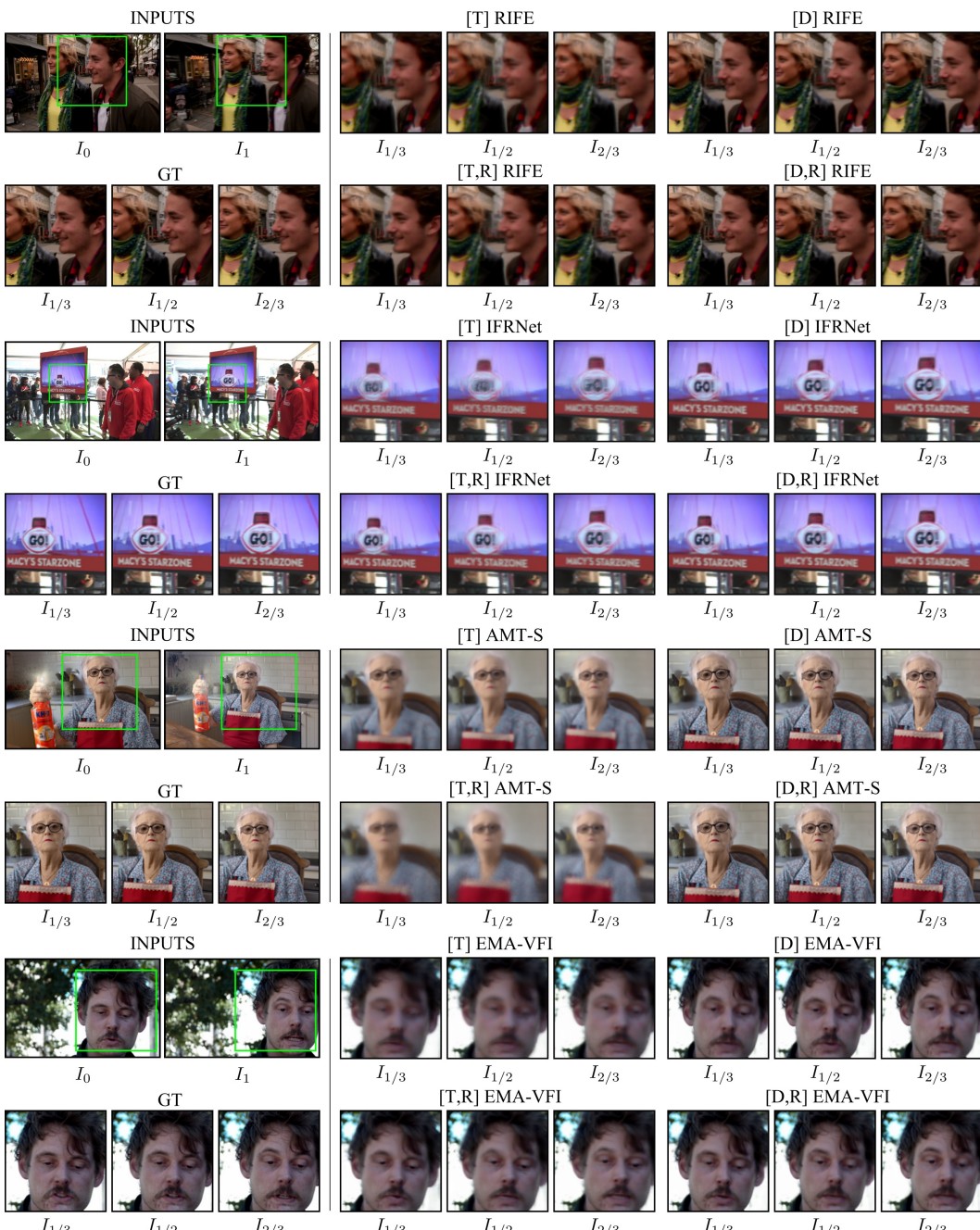

Figure 5: Comparison of qualitative results. **[T]**: original arbitrary time VFI models using time indexing. **[D]**: models using our distance indexing. **[T,R]**: models using time indexing with iterative reference-based estimation. **[D,R]**: models using both strategies. **Zoom in for a closer look**.

enhanced models $[D]$ and $[D, R]$, utilizing identical inputs as model $[T]$. Given the misalignment between predicted frames using a uniform distance map and the ground-truth, the enhanced models do not outperform the base model on pixel-centric metrics. However, we argue that in most applications, the goal of VFI is not to predict pixel-wise aligned frames, but to generate plausible frames with high perceptual quality. Furthermore, pixel-centric metrics are less sensitive to blur (Zhang et al., 2018), the major artifact introduced by velocity ambiguity. The pixel-centric metrics are thus less informative and denoted in gray. On perceptual metrics (especially NIQE), the enhanced models significantly outperforms the base model. This consistency with our qualitative observations further validates the effectiveness of distance indexing and iterative reference-based estimation.

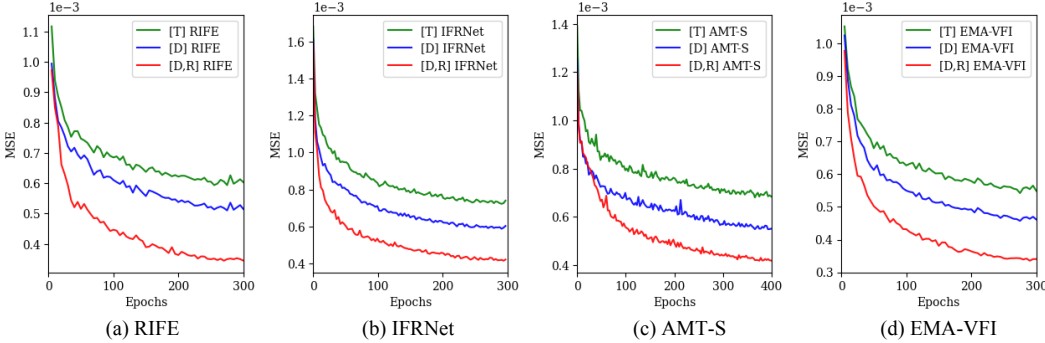

| (a) RIFE | (b) IFRNet | (c) AMT-S | (d) EMA-VFI |
|---|---|---|---|

Figure 6: Convergence curves. $[T]$ denotes traditional time indexing. $[D]$ denotes the proposed distance indexing. $[R]$ denotes iterative reference-based estimation.

Table 1: Comparison on Vimeo90K Septuplet dataset. $[T]$ denotes the method trained with traditional arbitrary time indexing paradigm. $[D]$ and $[R]$ denote the distance indexing paradigm and iterative reference-based estimation strategy, respectively. $[R]$ uses 2 iterations by default. $[\cdot]_u$ denotes inference with uniform map as time indexes. We utilize the first and last frames as inputs to predict the rest five frames. The **bold font** denotes the best performance in cases where comparison is possible. While the gray font indicates that the scores for pixel-centric metrics, PSNR and SSIM, are not calculated using strictly aligned ground-truth and predicted frames.

|  | RIFE Huang et al. (2022) | | | IFRNet Kong et al. (2022) | | | AMT-S Li et al. (2023) | | | EMA-VFI Zhang et al. (2023) | | |
|---|---|---|---|---|---|---|---|---|---|---|---|---|
|  | $[T]$ | $[D]$ | $[D,R]$ | $[T]$ | $[D]$ | $[D,R]$ | $[T]$ | $[D]$ | $[D,R]$ | $[T]$ | $[D]$ | $[D,R]$ |
| PSNR ↑ | 28.22 | **29.20** | 28.84 | 28.26 | **29.25** | 28.55 | 28.52 | **29.61** | 28.91 | 29.41 | **30.29** | 25.10 |
| SSIM ↑ | 0.912 | **0.929** | 0.926 | 0.915 | **0.931** | 0.925 | 0.920 | **0.937** | 0.931 | 0.928 | **0.942** | 0.858 |
| LPIPS ↓ | 0.105 | 0.092 | **0.081** | 0.088 | 0.080 | **0.072** | 0.101 | 0.086 | **0.077** | 0.086 | **0.078** | 0.079 |
| NIQE ↓ | 6.663 | 6.475 | **6.286** | 6.422 | 6.342 | **6.241** | 6.866 | 6.656 | **6.464** | 6.736 | 6.545 | **6.241** |
|  | $[T]$ | $[D]_u$ | $[D,R]_u$ | $[T]$ | $[D]_u$ | $[D,R]_u$ | $[T]$ | $[D]_u$ | $[D,R]_u$ | $[T]$ | $[D]_u$ | $[D,R]_u$ |
| PSNR ↑ | 28.22 | 27.55 | 27.41 | 28.26 | 27.40 | 27.13 | 28.52 | 27.33 | 27.17 | 29.41 | 28.24 | 24.73 |
| SSIM ↑ | 0.912 | 0.902 | 0.901 | 0.915 | 0.902 | 0.899 | 0.920 | 0.902 | 0.902 | 0.928 | 0.912 | 0.851 |
| LPIPS ↓ | 0.105 | 0.092 | **0.086** | 0.088 | 0.083 | **0.078** | 0.101 | 0.090 | **0.081** | 0.086 | **0.079** | 0.081 |
| NIQE ↓ | 6.663 | 6.344 | **6.220** | 6.422 | 6.196 | **6.167** | 6.866 | 6.452 | **6.326** | 6.736 | 6.457 | **6.227** |

Table 2: Ablation study of the number of iterations on Vimeo90K septuplet dataset. $[\cdot]^{\#}$ denotes the number of iterations used for inference.

|  | RIFE Huang et al. (2022) | | | IFRNet Kong et al. (2022) | | | AMT-S Li et al. (2023) | | | EMA-VFI Zhang et al. (2023) | | |
|---|---|---|---|---|---|---|---|---|---|---|---|---|
| $[D,R]_u$ | $[\cdot]^1$ | $[\cdot]^2$ | $[\cdot]^3$ | $[\cdot]^1$ | $[\cdot]^2$ | $[\cdot]^3$ | $[\cdot]^1$ | $[\cdot]^2$ | $[\cdot]^3$ | $[\cdot]^1$ | $[\cdot]^2$ | $[\cdot]^3$ |
| LPIPS ↓ | 0.093 | 0.086 | **0.085** | 0.085 | **0.078** | 0.078 | 0.086 | **0.081** | 0.081 | 0.084 | 0.081 | **0.080** |
| NIQE ↓ | 6.331 | 6.220 | **6.186** | 6.205 | 6.167 | **6.167** | 6.402 | **6.326** | 6.327 | 6.303 | 6.227 | **6.211** |
| $[T,R]$ | $[\cdot]^1$ | $[\cdot]^2$ | $[\cdot]^3$ | $[\cdot]^1$ | $[\cdot]^2$ | $[\cdot]^3$ | $[\cdot]^1$ | $[\cdot]^2$ | $[\cdot]^3$ | $[\cdot]^1$ | $[\cdot]^2$ | $[\cdot]^3$ |
| LPIPS ↓ | 0.103 | 0.087 | **0.087** | 0.091 | 0.084 | **0.084** | **0.106** | 0.135 | 0.157 | 0.088 | **0.083** | 0.085 |
| NIQE ↓ | 6.551 | 6.300 | **6.206** | 6.424 | 6.347 | **6.314** | **6.929** | 7.246 | 7.502 | 6.404 | 6.280 | **6.246** |

Lastly, Table 2 offers an ablation study on the number of iterations and the efficacy of a pure iterative reference-based estimation strategy. The upper section suggests that setting iterations at two strikes a good trade-off between computational efficiency and performance. The lower segment illustrates that while iterative reference-based estimation generally works for time indexing, there are exceptions, as observed with AMT-S. However, when combined with distance indexing, iterative reference-based estimation exhibits more stable improvement, as evidenced by the results for $[D,R]_u$. This is consistent with qualitative comparison. See more results in Appendix C.

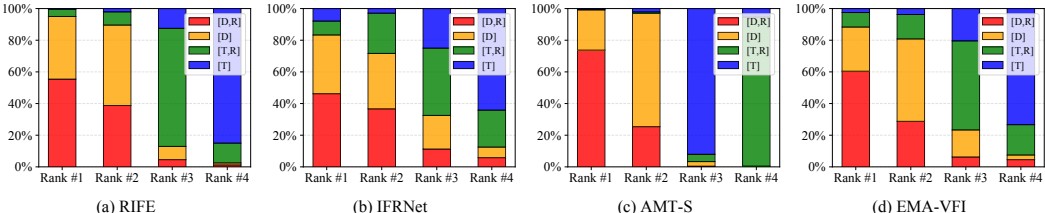

(a) RIFE          (b) IFRNet          (c) AMT-S          (d) EMA-VFI

Figure 7: User study. The horizontal axis represents user rankings, where #1 is the best and #4 is the worst. The vertical axis indicates the percentage of times each model variant received a specific ranking. Each model variant was ranked an equal number of times. The model $[D, R]$ emerged as the top performer in the study.

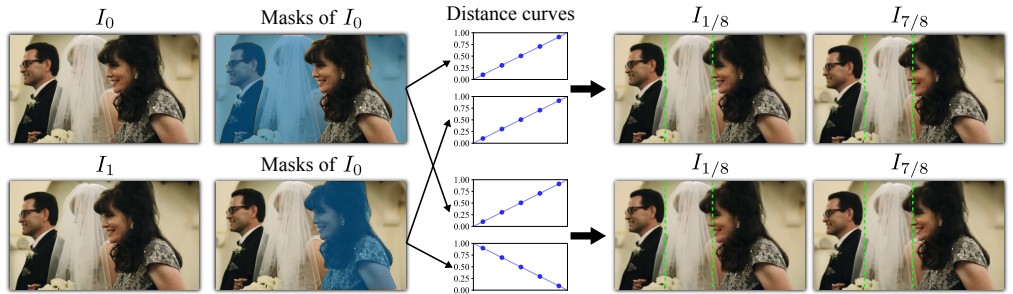

Figure 8: Manipulated interpolation of anything. Leveraging Segment-Anything (Kirillov et al., 2023), users can tailor distance curves for selected masks. Distinct masks combined with varying distance curves generate unique distance map sequences, leading to diverse interpolation outcomes.

## 5.4 USER STUDY

To validate the effectiveness of our proposed strategies, we further conducted a user study with 30 anonymous participants. Participants were tasked with ranking the interpolation quality of frames produced by four model variants: [T], [D], [T,R], and [D,R]. Please see details of user study UI in Appendix E. The results, presented in Figure 7, align with our qualitative and quantitative findings. The [D,R] model variant emerged as the top-rated, underscoring the effectiveness of our strategies.

## 5.5 2D MANIPULATION OF FRAME INTERPOLATION

Beyond simply enhancing the performance of VFI models, distance indexing equips them with a novel capability: tailoring the interpolation patterns for each individual object, termed as "manipulated interpolation of anything". Figure 8 demonstrates the workflow. The first stage employs SAM (Kirillov et al., 2023) to produce object masks for the starting frame. Users can then customize the distance curve for each object delineated by the mask, effectively controlling its interpolation pattern, *e.g.*, having one person moving backward in time. The backend of the application subsequently generates a sequence of distance maps based on these specified curves for interpolation. One of the primary applications is re-timing specific objects (**See the supplemetary video**).

## 6 CONCLUSION AND FUTURE WORK

We challenge the traditional time indexing paradigm and address its inherent uncertainties related to velocity distribution. Through the introduction of distance indexing and iterative reference-based estimation strategies, we offer a transformative paradigm to VFI. Our innovative plug-and-play strategies not only improves the performance in video interpolation but also empowers users with granular control over interpolation patterns across varied objects. Estimating accurate distance ratio maps from multiple frames represents a direction for our future research. Furthermore, the insights gleaned from our strategies have potential applications across a range of tasks that employ time indexing, such as space-time super-resolution, future predictions, blur interpolation and more.

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

## A    PROOF OF EQ. 4

Eq. 4 can be rigorously proven if an L2 loss is used,

$$\min_{\hat{I}_t} L = \mathbb{E}_{I_t \sim \mathcal{F}(I_0, I_1, t)}[(\hat{I}_t - I_t)^2]. \tag{11}$$

By setting the gradient to zero (this assumes during training, the neural network can reach the exact solution at this point),

$$
\begin{aligned}
\frac{\partial L}{\partial \hat{I}_t} &= 0 \\
\mathbb{E}_{I_t \sim \mathcal{F}(I_0, I_1, t)}\Big[\frac{\partial}{\partial \hat{I}_t}(\hat{I}_t - I_t)^2\Big] &= 0 \\
\mathbb{E}_{I_t \sim \mathcal{F}(I_0, I_1, t)}[2(\hat{I}_t - I_t)] &= 0 \\
\mathbb{E}_{I_t \sim \mathcal{F}(I_0, I_1, t)}[\hat{I}_t] - \mathbb{E}_{I_t \sim \mathcal{F}(I_0, I_1, t)}[I_t] &= 0 \\
\hat{I}_t &= \mathbb{E}_{I_t \sim \mathcal{F}(I_0, I_1, t)}[I_t]
\end{aligned}
\tag{12}
$$

For other losses, this specific equation no longer holds, but we empirically observe that the model still learns an aggregated mixture of training frames which results in blur (RIFE (Huang et al., 2022) and EMA-VFI (Zhang et al., 2023): Laplacian loss, *i.e.*, L1 loss between the Laplacian pyramids of image pairs; IFRNet (Kong et al., 2022) and AMT (Li et al., 2023): Charbonnier loss).

## B    THE RATIONALE FOR SOLVING AMBIGUITY

First and foremost, it is essential to clarify that **velocity ambiguity can solely exist and be resolved in the training phase, not in the inference phase**. The key idea behind our approach can be summarized as follows: While conventional VFI methods with time indexing rely on a one-to-many mapping, our distance indexing learns an approximate one-to-one mapping, which resolves the ambiguity during training. When the input-output relationship is one-to-many during training, the training process fluctuates among conflicting objectives, ultimately preventing convergence towards any specific optimization goal. In VFI, the evidence is the generation of blurry images in the inference phase. Once the ambiguity has been resolved using the new indexing method in the training phase, the model can produce significantly clearer results regardless of the inference strategy used.

Indeed, this one-to-many ambiguity in training is not unique to VFI, but for a wide range of machine learning problems. In some areas, researchers have come up with similar methods.

**A specific instantiation of this problem:**    Let us look at an example in text-to-speech (TTS). The same text can be paired with a variety of speeches, and direct training without addressing ambiguities can result in a **"blurred" voice** (a statistical average voice). To mitigate this, a common approach is to incorporate a speaker embedding vector or a style embedding vector (representing different gender, accents, speaking styles, etc.) during training, which helps reduce ambiguity. **During the inference phase, utilizing an average user embedding vector can yield high-quality speech output.** Furthermore, by manipulating the speaker embedding vector, effects such as altering the accent and pitch can also be achieved.

Here is a snippet from a high-impact paper (Wang et al., 2018) which came up with the style embedding in TTS:

> Many TTS models, including recent end-to-end systems, only learn an averaged prosodic distribution over their input data, generating less expressive speech – especially for long-form phrases. Furthermore, they often lack the ability to control the expression with which speech is synthesized.

Understanding this example can significantly help understand our paper, as there are many similarities between the two, *e.g.*, motivation, solution, and manipulation.

**A minimal symbolic example to help understand better:** Assuming we want to train a mapping function $\mathcal{F}$ from numbers to characters.

**Training input-output pairs with ambiguity** ($\mathcal{F}$ is optimized):

$$1 \xrightarrow{\mathcal{F}} a, 1 \xrightarrow{\mathcal{F}} b, 2 \xrightarrow{\mathcal{F}} a, 2 \xrightarrow{\mathcal{F}} b$$

$\mathcal{F}$ is optimized with some losses involving the input-output pairs above:

$$\min_{\mathcal{F}} \quad L(\mathcal{F}(1), a) + L(\mathcal{F}(1), b) + L(\mathcal{F}(2), a) + L(\mathcal{F}(2), b),$$

where $L$ can be L1, L2 or any other kind of losses. Because the same input is paired with multiple different outputs, the model $\mathcal{F}$ is optimized to learn an average (or, generally, a mixture) of the conflicting outputs, which results in blur at inference.

Inference phase ($\mathcal{F}$ is fixed):

$$1 \xrightarrow{\mathcal{F}} \{a, b\}?, 2 \xrightarrow{\mathcal{F}} \{a, b\}?$$

**Training without ambiguity** ($\mathcal{F}$ is optimized):

$$1 \xrightarrow{\mathcal{F}} a, 1 \xrightarrow{\mathcal{F}} a, 2 \xrightarrow{\mathcal{F}} b, 2 \xrightarrow{\mathcal{F}} b$$

In the input-output pairs above, each input value is paired with exactly one output value. Therefore, $\mathcal{F}$ is trained to learn a unique and deterministic mapping.

Inference phase ($\mathcal{F}$ is fixed):

$$1 \xrightarrow{\mathcal{F}} a, 2 \xrightarrow{\mathcal{F}} b$$

**Coming back to VFI:** When time indexing is used, the same $t$ value is paired to images where the objects are located at various locations due to the speed and directional ambiguities. When distance mapping is used, a single $d$ value is paired to images where the objects are always at the same distance ratio, which allows the model to learn a more deterministic mapping for resolving the speed ambiguity.

It is important to note that fixing the ambiguity does not solve all the problems: At inference time, the "correct" (close to ground-truth) distance map is not available. In this work, we show that it is possible to provide uniform distance maps as inputs to generate a clear output video, which is not perfectly pixel-wise aligned with the ground truth. This is the reason why the proposed method does not achieve state-of-the-art in terms of PSNR and SSIM in Tab. 1. However, it achieves sharper frames with higher perceptual quality, which is shown by the better LPIPS and NIQE.

We claim the "correct" distance map is hard to estimate accurately from merely two frames since there are a wide range of possible velocities. If considering more neighboring frames like Xu et al. (2019) (more observation information), it is possible to estimate an accurate distance map for pixel-wise aligned interpolation, which we leave for future work.

Furthermore, manipulating distance maps corresponds to sampling other possible unseen velocities, *i.e.*, 2D manipulation of frame interpolation, similar to that mentioned TTS paper (Wang et al., 2018).

## C  ADDITIONAL EXPERIMENTS

**Comparison of the fixed-time setting:** Using RIFE (Huang et al., 2022) as a representative example, we extend our comparison to the fixed-time training paradigm, depicted in Figure 9. The label $[T]$ RIFE (Tri) refers to the model trained on the Triplet dataset from Vimeo90K (Xue et al., 2019) employing time indexing. Conversely, $[D]$ RIFE (Tri) indicates training on the same Triplet dataset but utilizing our distance indexing approach. Both $[D]$ RIFE and $[D, R]$ RIFE models are trained on the septuplet dataset, consistent with our earlier comparison. Despite being trained on varied datasets, it is evident that the arbitrary time model outperforms the fixed time model. However, the efficacy of distance indexing appears restrained within the fixed-time training paradigm. This limitation stems from the fact that deriving distance representation solely from the middle frame yields a sparse distribution, making it challenging for the network to grasp the nuances of distance.

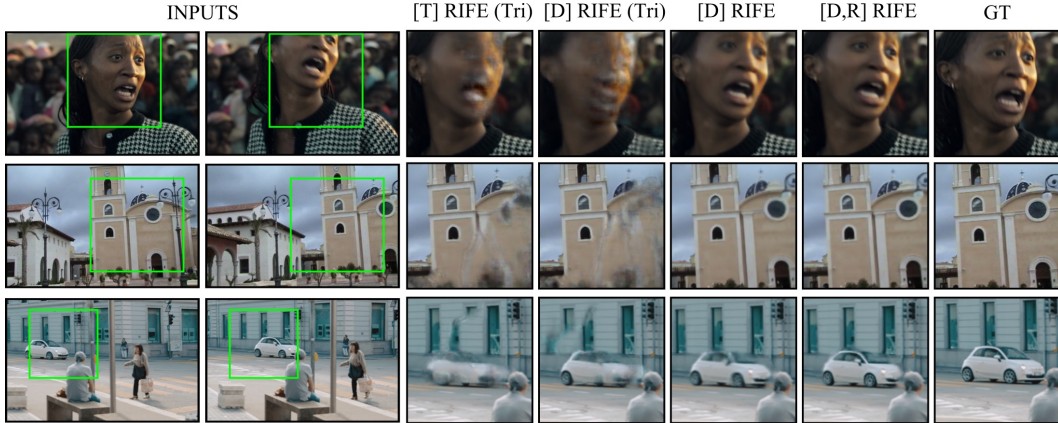

Figure 9: Additional comparison of qualitative results. [T] RIFE (Tri) denotes RIFE (Huang et al., 2022) trained in a fixed time indexing paradigm (Vimeo90K triplet dataset (Xue et al., 2019)). [D] RIFE (Tri) denotes the corresponding model trained using distance indexing.

Table 3: Comparison on Vimeo90K Triplet dataset. $[T]$ denotes the method trained with traditional fixed time indexing paradigm. $[D]$ denotes the distance indexing paradigm. $[\cdot]_u$ denotes inference with uniform map as time indexes.

|  | RIFE Huang et al. (2022) | | | IFRNet Kong et al. (2022) | | | AMT-S Li et al. (2023) | | | EMA-VFI Zhang et al. (2023) | | |
| --- | --- | --- | --- | --- | --- | --- | --- | --- | --- | --- | --- | --- |
|  | $[T]$ | $[D]$ | $[D]_u$ | $[T]$ | $[D]$ | $[D]_u$ | $[T]$ | $[D]$ | $[D]_u$ | $[T]$ | $[D]$ | $[D]_u$ |
| PSNR ↑ | 35.61 | **36.04** | 35.18 | 35.80 | **36.26** | 35.14 | 35.97 | **36.56** | 35.21 | 36.50 | **37.13** | 36.21 |
| SSIM ↑ | 0.978 | **0.979** | 0.976 | 0.979 | **0.981** | 0.977 | 0.980 | **0.982** | 0.977 | 0.982 | **0.983** | 0.981 |
| LPIPS ↓ | 0.022 | **0.022** | 0.023 | 0.020 | **0.019** | 0.021 | 0.021 | **0.020** | 0.023 | 0.020 | **0.019** | 0.020 |
| NIQE ↓ | 5.249 | 5.225 | **5.224** | 5.256 | 5.245 | **5.225** | 5.308 | 5.293 | **5.288** | 5.372 | 5.343 | **5.335** |

We delve deeper into the quantitative analysis of these findings in Table 3. Compared to training arbitrary time models on the Septuplet dataset, the advantages of distance indexing become notably decreased when training fixed time models on the Triplet dataset.

**Comparison on other benchmarks:**   The septuplet set of Vimeo90K (Xue et al., 2019) is large enough to train a practical video frame interpolation model, and it represents the situations where the temporal distance between input frames is large. Thus, Vimeo90K (septuplet) can best demonstrate the velocity ambiguity problem that our work aims to highlight. Nonetheless, we report more results on other benchmarks. Table 4 and Table 5 show the results of RIFE (Huang et al., 2022) on Adobe240 (Su et al., 2017) and X4K1000FPS (Sim et al., 2021) for ×8 interpolation respectively, using uniform maps. Distance indexing $[D]_u$ and iterative reference-based estimation $[R]_u$ strategies can consistently help improve the perceptual quality. In addition, it is noteworthy that $[D]_u$ is better than $[T]$ in terms of the pixel-centric metrics like PSNR, showing that the constant speed assumption (uniform distance maps) holds well on these two easier benchmarks. We further show ×16 interpolation on X4K1000FPS with larger temporal distance in Table 6. The results highlight that the benefits of our strategies are more pronounced with increased temporal distances.

**Comparison of using other optical flow estimator:**   We also employ GMFlow (Xu et al., 2022) for the precomputation of distance maps, enabling an analysis of model performance when integrated with alternative optical flow estimations. The results are as shown in Table 7. Our strategies still lead to consistent improvement on perceptual metrics. However, this more recent and performant optical flow estimator does not introduce improvement compared to RAFT (Teed & Deng, 2020). A likely explanation is that since we quantify the optical flow to $[0, 1]$ scalar values for better generalization, our training strategies are less sensitive to the precision of the optical flow estimator.

Table 4: Comparison on Adobe240 (Su et al., 2017) for ×8 interpolation. We use RIFE (Huang et al., 2022) as a representative example.

|  | $[T]$ | $[D]_u$ | $[D, R]_u$ |
|---|---|---|---|
| PSNR ↑ | 30.24 | **30.47** | 30.30 |
| SSIM ↑ | **0.939** | 0.938 | 0.937 |
| LPIPS ↓ | 0.073 | 0.057 | **0.054** |
| NIQE ↓ | 5.206 | 4.974 | **4.907** |

Table 5: Comparison on X4K1000FPS (Sim et al., 2021) for ×8 interpolation. We use RIFE (Huang et al., 2022) as a representative example.

|  | $[T]$ | $[D]_u$ | $[D, R]_u$ |
|---|---|---|---|
| PSNR ↑ | 36.36 | **36.80** | 36.26 |
| SSIM ↑ | **0.967** | 0.964 | 0.964 |
| LPIPS ↓ | 0.040 | 0.032 | **0.032** |
| NIQE ↓ | 7.130 | 6.936 | **6.924** |

Table 6: Comparison on X4K1000FPS (Sim et al., 2021) for ×16 interpolation. We use RIFE (Huang et al., 2022) as a representative example.

|  | $[T]$ | $[D]_u$ | $[D, R]_u$ |
|---|---|---|---|
| PSNR ↑ | 31.04 | **31.60** | 31.52 |
| SSIM ↑ | 0.910 | 0.914 | **0.922** |
| LPIPS ↓ | 0.104 | 0.094 | **0.079** |
| NIQE ↓ | 7.215 | 6.953 | **6.927** |

Table 7: Comparison on the septuplet of Vimeo90K (Xue et al., 2019) using GMFlow (Xu et al., 2022) for distance map calculation. We use RIFE (Huang et al., 2022) as a representative example.

|  | $[T]$ | $[D]_u$ | $[D, R]_u$ |
|---|---|---|---|
| PSNR ↑ | **28.22** | 27.29 | 26.96 |
| SSIM ↑ | **0.912** | 0.898 | 0.895 |
| LPIPS ↓ | 0.105 | 0.101 | **0.092** |
| NIQE ↓ | 6.663 | 6.449 | **6.280** |

Table 8: Comparison on the septuplet of Vimeo90K (Xue et al., 2019) using LPIPS loss (Zhang et al., 2018). We use RIFE (Huang et al., 2022) as a representative example.

|  | $[T]$ | $[D]_u$ | $[D, R]_u$ |
|---|---|---|---|
| PSNR ↑ | **27.19** | 26.71 | 26.72 |
| SSIM ↑ | **0.898** | 0.889 | 0.890 |
| LPIPS ↓ | **0.061** | 0.065 | 0.064 |
| NIQE ↓ | 6.307 | 5.901 | **5.837** |

**Comparison of using perceptual loss:** We present the results of employing the more recent LPIPS loss (Zhang et al., 2018) with a VGG backbone (Simonyan & Zisserman, 2014), as shown in Table 8. The non-reference perceptual quality metric, NIQE, shows notable improvement across all variants. The results also consistently demonstrate the effectiveness of our strategies in resolving velocity ambiguity. Besides, due to the direct optimization of LPIPS loss, the assumption of constant speed in uniform maps affects the performance for this metric.

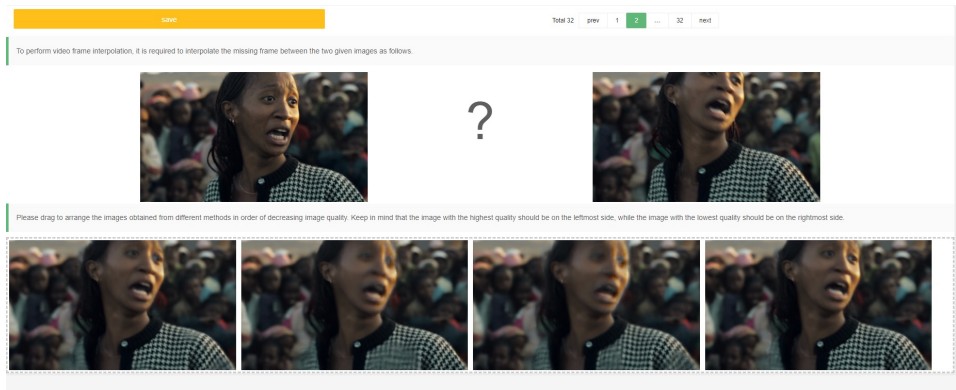

Figure 10: User interface of user study.

## D  COSTS OF PROPOSED STRATEGIES

**Distance indexing:** Transitioning from time indexing ($[T]$) to distance indexing ($[D]$) does not introduce extra computational costs during the inference phase, yet significantly enhancing image quality. In the training phase, the primary requirement is a one-time computation (offline) of distance maps for each image triplet.

**Iterative reference-based estimation** ($[D, R]$)**:** Given that the computational overhead of merely expanding the input channel, while keeping the rest of the structure unchanged, is negligible, the computational burden during the training phase remains equivalent to that of the $[D]$ model. While during inference, the total consumption is equal to the number of iterations $\times$ the consumption of the $[D]$ model. We would like to highlight that this iterative strategy is optional: Users can adopt this strategy at will when optimal interpolation results are demanded and the computational budget allows. Our experiments in Table 2 show that 2 iterations, *i.e.*, doubling the computational cost, yield cost-effective improvements.

## E  USER STUDY UI

As shown in Figure 10, we initially presented users with the input starting and ending frames. Subsequently, the results from each model's four distinct variants were displayed anonymously in a sequence, with the order shuffled for each presentation. Users were tasked with reordering the images by dragging them, placing them from left to right based on their perceived quality, *i.e.*, the best image on the extreme left and the least preferred on the far right.

## F  DEMO

We have included a video demo (available in the supplementary materials as "supp.mp4") to intuitively showcase the enhanced quality achieved through our strategies. The video further illustrates the idea of manipulating object interpolations and provides a guide on using the related web application.

## G  CODE

We further include source code and detailed instructions in the supplementary material to support the reproducibility of our work.

## H  LIMITATIONS

Despite the promising advancements introduced by our strategies, there are a few inherent limitations. Firstly, the current system only allows for manipulations along a fixed trajectory. It does not

possess the capability to model the distribution of possible trajectories. Thus, exploring methods to recover any potential direction, or perhaps to learn a distribution of directions, is a direction for our future research. In our current approach, we use a uniform distance map which assumes every object moves at constant speed along their trajectories. In practice, different objects may have different acceleration and therefore this constant speed assumption may lead to sub-optimal frame interpolation. As discussed in Section B, one potential future direction is to use multiple consecutive frames to capture the different acceleration of each object, which is then used to compute the precise distance ratio for each of them. This may further boost the performance and even beat time indexing methods on pixel-wise metrics during inference. Additionally, the overall effectiveness and accuracy of the model are intrinsically tied to the precision of optical flow estimation and the quality of segmentation masks (optional). Any inaccuracies or shortcomings in these preliminary steps can cascade and potentially impact the final interpolation results.

