# OpenReview forum: "Clearer Frames, Anytime: Resolving Velocity Ambiguity in Video Frame Interpolation"
_ICLR.cc/2024/Conference — ICLR 2024 Conference Desk Rejected Submission_

### Official Review · Reviewer_BPE4 · 2023-10-28

**Soundness:** 3 good
**Presentation:** 3 good
**Contribution:** 3 good
**Rating:** 8
**Confidence:** 2

**Summary:**

This paper presents an approach to Video Frame Interpolation (VFI) termed "distance indexing," aiming to improve the precision of object movements in interpolated frames. Instead of previous "time indexing," the proposed method gives the network explicit hints about the distance an object has traveled between the start and end frames, reducing the uncertainty tied to object speeds. To address directional ambiguity in long-range motion, the paper also introduces an iterative reference-based estimation strategy that breaks down a long-range prediction into several short-range steps.

**Strengths:**

- motivation is clear.
- good performance. By employing "distance indexing," the paper significantly improves the precision of object movements in interpolated frames over traditional "time indexing" methods. Since the training strategy can be played as a plug-and-play strategy, it can be extended to another video frame interpolate methods.
- The proposed method provides some analysis for some possible reasons to cause blur in video frame interpolate task.

**Weaknesses:**

- The training and inference time complexity will sightly increasing due to introducing the proposed tech.

**Questions:**

NA

---

> ### Author Response · Authors · 2023-11-14
>
> Thank you for recognizing our work's motivation, performance, and interpretability.
>
> Before addressing specific questions, if there are any uncertainties or if a refresher on the core concepts of our paper's methodology is needed, we kindly invite you to review our comprehensive response to Reviewer aknz, where we offer a clearer and more detailed explanation:
>
> `First and foremost, it's essential to clarify that ... similar to that mentioned TTS paper (Wang, Yuxuan, et al.).`
>
>
>
> > The training and inference time complexity will sightly increasing due to introducing the proposed tech.
>
> We will incorporate the following details into our final paper:
>
> **Distance indexing**: Transitioning from time indexing ($[T]$) to distance indexing ($[D]$) does not introduce extra computational costs during the inference phase, yet significantly enhancing image quality. In the training phase, the primary requirement is a one-time computation (offline) of distance maps for each image triplet.
>
> **Iterative reference-based estimation ($[D,R]$)**: Given that the computational overhead of merely expanding the input channel, while keeping the rest of the structure unchanged, is negligible, the computational burden during the training phase remains equivalent to that of the $[D]$ model. While during inference, the total consumption is equal to the number of iterations $\times$ the consumption of the $[D]$ model. We would like to highlight that this iterative strategy is optional: Users can adopt this strategy at will when optimal interpolation results are demanded and the computational budget allows. Our experiments (Table 2) show that 2 iterations, *i.e.,* doubling the computational cost, yield cost-effective improvements.
>
>
>
> Should you have any further questions about our research, including those raised by other reviewers, we are more than happy to engage in further discussion and provide explanations!

---

### Official Review · Reviewer_PtYf · 2023-10-31

**Soundness:** 3 good
**Presentation:** 3 good
**Contribution:** 3 good
**Rating:** 6
**Confidence:** 5

**Summary:**

This paper presents a novel and controllable approach for video frame interpolation. In contrast to existing VFI models that typically rely on time indexing to generate intermediate frames, the authors introduce a distance indexing strategy to address the problem of velocity ambiguity. Additionally, they incorporate an iterative reference-based estimation method to improve the quality of the interpolated frames. The proposed method can be easily integrated into existing VFI models as a plug-and-play solution. As a result, this work not only enhances the performance of VFI models but also provides a new video editing tool.

**Strengths:**

1. This paper presents an intriguing and intuitively effective solution for video frame interpolation.
2. The authors have conducted extensive experiments to thoroughly evaluate the effectiveness of the proposed method.
3. The paper is well-written and provides sufficient explanations of the details, making it easy to follow.

**Weaknesses:**

1. It would be even more beneficial to evaluate the proposed method on other high-resolution benchmarks for testing, particularly to demonstrate its superiority in handling larger motion scenarios.
2. Additionally, please specify which three out of the seven frames were used for testing purposes.
3. I am also interested in exploring the robustness of the proposed method when dealing with inaccurate optical flow estimation. Additionally, I am curious to see how well the model performs when incorporating it with another optical flow estimator, such as FlowNet, to provide the distance ratio map.
4. Provide a comparison of the inference cost between different methods, including additional optical flow estimation and iterative reference-based interpolation. It is important to note that the iterative reference-based strategy may involve running the interpolation multiple times, resulting in a significant increase in the overall inference time.

**Questions:**

1. Evaluation of the proposed method on benchmarks specifically designed for high-resolution interpolation.
2. Analysis of the robustness of the method in scenarios where the optical flow estimation is inaccurate.

---

> ### Author Response · Authors · 2023-11-14
>
> Thanks for recognizing the effectiveness of our approach and for the constructive suggestions. We will first address the concerns raised, except those requiring experimental validation.
>
> Before addressing specific questions, if there are any uncertainties or if a refresher on the core concepts of our paper's methodology is needed, we kindly invite you to review our comprehensive response to Reviewer aknz, where we offer a clearer and more detailed explanation:
>
> `First and foremost, it's essential to clarify that ... similar to that mentioned TTS paper (Wang, Yuxuan, et al.).`
>
>
>
> > Additionally, please specify which three out of the seven frames were used for testing purposes.
>
> Notice that, within each septuplet, the i-th frame corresponds to $I_{i/6}$ in Fig. 5 ($i=0,1,\cdots,6$).
> We utilized the first ($I_0$) and last ($I_1$) frames as inputs to predict the rest five frames. We only show $I_{2/6},I_{3/6},I_{4/6}$ in Fig. 5 for visualization, but we report the results for all five intermediate frames in quantitative results.
> This detail will be explicitly clarified in the final version of our paper.
>
>
>
> > Provide a comparison of the inference cost between different methods, including additional optical flow estimation and iterative reference-based interpolation. It is important to note that the iterative reference-based strategy may involve running the interpolation multiple times, resulting in a significant increase in the overall inference time.
>
> **Distance indexing**: Transitioning from time indexing ($[T]$) to distance indexing ($[D]$) does not introduce extra computational costs during the inference phase, yet significantly enhancing image quality. In the training phase, the primary requirement is a one-time computation (offline) of distance maps for each image triplet.
>
> **Iterative reference-based estimation ($[D,R]$)**: Given that the computational overhead of merely expanding the input channel, while keeping the rest of the structure unchanged, is negligible, the computational burden during the training phase remains equivalent to that of the $[D]$ model. While during inference, the total consumption is equal to the number of iterations $\times$ the consumption of the $[D]$ model. We would like to highlight that this iterative strategy is optional: Users can adopt this strategy at will when optimal interpolation results are demanded and the computational budget allows. Our experiments (Table 2) show that 2 iterations, *i.e.,* doubling the computational cost, yield cost-effective improvements.
>
> We will include these details in our final paper for clarity.

---

> ### Author Response · Authors · 2023-11-18
>
> > It would be even more beneficial to evaluate the proposed method on other high-resolution benchmarks for testing, particularly to demonstrate its superiority in handling larger motion scenarios.
>
> Thank you for your constructive suggestions! Using RIFE as an example, we present experimental comparisons for 8x interpolation on the Adobe240 [1] and X4K1000FPS [2] benchmarks with uniform maps. The results are provided below for reference:
>
> | Adobe240 (x8)   | PSNR    | SSIM   | LPIPS  | NIQE   |
> | ------- | ------- | ------ | ------ | ------ |
> | RIFE [T]        | 30.2430 | 0.9386 | 0.0729 | 5.2060 |
> | RIFE [D]$_{u}$  | 30.4704 | 0.9384 | 0.0565 | 4.9735 |
> | RIFE [D,R]$_{u}$ | 30.2961 | 0.9372 | 0.0542 | 4.9068 |
>
> | X4K1000FPS (x8) | PSNR    | SSIM   | LPIPS  | NIQE   |
> | ----- | ------- | ------ | ------ | ------ |
> | RIFE [T]        | 36.3557 | 0.9667 | 0.0403 | 7.1302 |
> | RIFE [D]$_{u}$  | 36.7964 | 0.9642 | 0.0315 | 6.9364 |
> | RIFE [D,R]$_{u}$ | 36.2588 | 0.9642 | 0.0315 | 6.9242 |
>
> Distance indexing [D] and iterative reference-based estimation [R] strategies can consistently help improve the perceptual quality. In addition, it is noteworthy that [D]$_u$ is better than [T]$_u$ in terms of the pixel-centric metrics like PSNR, showing that the constant speed assumption (uniform distance maps) holds well on these two easier benchmarks.
>
> We also show 16x interpolation on X4K1000FPS with larger temporal distance as follows:
>
> | X4K1000FPS (x16) | PSNR    | SSIM   | LPIPS  | NIQE   |
> | -------- | ------- | ------ | ------ | ------ |
> | RIFE [T]         | 31.0401 | 0.9102 | 0.1043 | 7.2152 |
> | RIFE [D]$_u$   | 31.6000 | 0.9141 | 0.0944 | 6.9531 |
> | RIFE [D,R]$_u$  | 31.5195 | 0.9220 | 0.0786 | 6.9265 |
>
> The results highlight that the benefits of our strategies are more pronounced with increased temporal distances.
>
> > I am also interested in exploring the robustness of the proposed method when dealing with inaccurate optical flow estimation. Additionally, I am curious to see how well the model performs when incorporating it with another optical flow estimator, such as FlowNet, to provide the distance ratio map.
>
> Since the inference speed of FlowNet is too slow, we use GMFlow [3], a newer optical flow estimator, to precompute distance maps for comparison. The results are as follows:
>
> | Vimeo90K (GMFlow) | PSNR    | SSIM   | LPIPS  | NIQE   |
> | ------- | ------- | ------ | ------ | ------ |
> | RIFE [T]          | 28.2175 | 0.9116 | 0.1048 | 6.6626 |
> | RIFE [D]$_u$      | 27.2927 | 0.8983 | 0.1009 | 6.4492 |
> | RIFE [D,R]$_u$     | 26.9583 | 0.8952 | 0.0915 | 6.2799 |
>
> Our strategies still lead to consistent improvement on perceptual metrics. However, this more recent and performant optical flow estimator does not introduce improvement compared to RAFT. A likely explanation is that since we quantify the optical flow to $[0,1]$ scalar values for better generalization, our training strategies are less sensitive to the precision of the optical flow estimator.
>
> P.S., the same experimental conclusions can be derived from other SOTA models like AMT as follows:
>
> | Adobe240 (x8) | PSNR    | SSIM   | LPIPS  | NIQE   |
> | ------ | ------- | ------ | ------ | ------ |
> | AMT [T]   | 29.5985 | 0.9333 | 0.0935 | 5.4004 |
> | AMT [D]$_u$  | 29.7449 | 0.9341 | 0.0700 | 5.0037 |
> | AMT [D,R]$_u$  | 29.4671 | 0.9329 | 0.0630 | 4.9392 |
>
> | X4K1000FPS (x8) | PSNR    | SSIM   | LPIPS  | NIQE   |
> | ------- | ------- | ------ | ------ | ------ |
> | AMT [T]         | 35.6691 | 0.9612 | 0.0598 | 7.3296 |
> | AMT [D]$_u$    | 36.6280 | 0.9631 | 0.0412 | 7.0355 |
> | AMT [D,R]$_u$   | 36.3044 | 0.9645 | 0.0355 | 6.8704 |
>
> | X4K1000FPS (x16) | PSNR    | SSIM   | LPIPS  | NIQE   |
> | ------- | ------- | ------ | ------ | ------ |
> | AMT [T]       | 31.6443 | 0.9242 | 0.1222 | 7.6546 |
> | AMT [D]$_u$   | 32.5098 | 0.9352 | 0.0816 | 7.1798 |
> | AMT [D,R]$_u$    | 31.9543 | 0.9311 | 0.0771 | 7.0707 |
>
> | Vimeo90k (GMFlow) | PSNR    | SSIM   | LPIPS  | NIQE   |
> | ------ | ------- | ------ | ------ | ------ |
> | AMT-S [T]         | 28.5202 | 0.9196 | 0.1011 | 6.8659 |
> | AMT-S [D]$_u$     | 27.0252 | 0.9004 | 0.1006 | 6.6257 |
> | AMT-S [D,R]$_u$    | 26.5586 | 0.8928 | 0.0937 | 6.4910 |
>
> We will incorporate the above discussions into the final paper.
>
> References:
>
> [1] Su, Shuochen, Mauricio Delbracio, Jue Wang, Guillermo Sapiro, Wolfgang Heidrich, and Oliver Wang. "Deep video deblurring for hand-held cameras." In *Proceedings of the IEEE conference on computer vision and pattern recognition*, pp. 1279-1288. 2017.
>
> [2] Sim, Hyeonjun, Jihyong Oh, and Munchurl Kim. "Xvfi: extreme video frame interpolation." In *Proceedings of the IEEE/CVF international conference on computer vision*, pp. 14489-14498. 2021.
>
> [3] Xu, Haofei, Jing Zhang, Jianfei Cai, Hamid Rezatofighi, and Dacheng Tao. "Gmflow: Learning optical flow via global matching." In *Proceedings of the IEEE/CVF conference on computer vision and pattern recognition*, pp. 8121-8130. 2022.

---

### Official Review · Reviewer_1WoD · 2023-11-01

**Soundness:** 4 excellent
**Presentation:** 4 excellent
**Contribution:** 4 excellent
**Rating:** 8
**Confidence:** 4

**Summary:**

Existing time indexing-based VFI methods suffer from the speed ambiguity during training, and the authors propose to instead use a distance indexing training mechanism to reduce the inconsistency of the acceleration of each objects, which may damage network training. In addition to address the direction ambiguity, the authors propose an iterative estimation mechanism. The authors conducted experiments on four state-of-the-art methods to validate the significant enhancement that the proposed plug-and-play modules bring to VFI. In addition, the authors show an exciting VFI DEMO that implements motion regions and trajectories editable in conjunction with SAM.

**Strengths:**

1. The motivation of why we need distance indexing for VFI tasks and the details of the two proposed modules are well presented, and the enhancements they bring to the existing VFI models are well illustrated.
2. The VFI DEMO that implements motion regions and trajectories editable in conjunction with SAM, which is very exciting.

**Weaknesses:**

1. The proposed distance indexing resolves the scalar velocity ambiguity in training, but nor in inference. Although the authors show in the experiments (Fig. 9) that the results achieved by using only uniform map in inference can already be satisfactory for humans, I think it would be better to explain this further in the Methods section.
2. Does the iterative estimation mechanism require any changes to the structure of the existing model, especially the input and output parts? If so, is this consistent with the plug-and-play claim? The details need to be clarified.
3. The training details of iterative estimation need to be clarified as well. Is D_{t/2} obtained in training using the optical flow computed from the ground-truth I_{t/2} in the dataset? Is it a uniform map of D=t/2 in inference?
4. The additional consumptions that the proposed new mechanisms bring to model training and inference need to be discussed.

**Questions:**

1. Does iterative estimation cause problems with error accumulation, e.g. the first time the direction is wrong, making it harder to get back to the right one the second time? Failure cases may be needed.
2. Is manipulable interpolation of optical flow based VFI methods also possible?
3. Would changing to a newer optical flow estimation method result in more improvements? Especially for the exception AMT-S.

---

> ### Author Response · Authors · 2023-11-14
>
> Thank you for acknowledging our contributions and for constructive feedback. We will first address the concerns raised, except those requiring experimental validation.
>
> > The proposed distance indexing resolves the scalar velocity ambiguity in training, but nor in inference. Although the authors show in the experiments (Fig. 9) that the results achieved by using only uniform map in inference can already be satisfactory for humans, I think it would be better to explain this further in the Methods section.
>
> Thank you for raising this concern. We kindly invite Reviewer 1WoD to refer to our detailed response to Reviewer aknz, where we present a more intuitive and detailed explanation:
>
> `First and foremost, it's essential to clarify that ... similar to that mentioned TTS paper (Wang, Yuxuan, et al.).`
>
>
>
> > Does the iterative estimation mechanism require any changes to the structure of the existing model, especially the input and output parts? If so, is this consistent with the plug-and-play claim? The details need to be clarified.
>
> The adaptation required is merely an expansion of the input channel for each model to include additional reference inputs, while the output configuration remains the same. More precisely, the iterative reference-based estimation mechanism increases the original 1-dimensional $t$ map channel to a 5-dimensional format. This includes a 1-dimensional $D_t$ map, a 1-dimensional $D_{ref}$ map, and a 3-dimensional RGB reference $I_{ref}$. Since this modification does not affect the core architecture of the model, we maintain that it aligns with the plug-and-play principle.
>
>
>
> > The training details of iterative estimation need to be clarified as well. Is D_{t/2} obtained in training using the optical flow computed from the ground-truth I_{t/2} in the dataset? Is it a uniform map of D=t/2 in inference?
>
> Thank you for your suggestion. Indeed, during training, $D_{ref}$ is derived using optical flow computed from the ground-truth at a time point corresponding to a randomly chosen reference frame, such as $t/2$. In the inference phase, we utilize a uniform map with $D=t/2$. We will ensure to clarify these details in the final version of our paper.
>
>
>
> >  The additional consumptions that the proposed new mechanisms bring to model training and inference need to be discussed.
>
> **Distance indexing**: Transitioning from time indexing ($[T]$) to distance indexing ($[D]$) does not introduce extra computational costs during the inference phase, yet significantly enhancing image quality. In the training phase, the primary requirement is a one-time computation (offline) of distance maps for each image triplet.
>
> **Iterative reference-based estimation ($[D,R]$)**: Given that the computational overhead of merely expanding the input channel, while keeping the rest of the structure unchanged, is negligible, the computational burden during the training phase remains equivalent to that of the $[D]$ model. While during inference, the total consumption is equal to the number of iterations $\times$ the consumption of the $[D]$ model. We would like to highlight that this iterative strategy is optional: Users can adopt this strategy at will when optimal interpolation results are demanded and the computational budget allows. Our experiments (Table 2) show that 2 iterations, *i.e.,* doubling the computational cost, yield cost-effective improvements. We will include these details in our final paper for clarity.
>
>
>
> > Does iterative estimation cause problems with error accumulation, e.g. the first time the direction is wrong, making it harder to get back to the right one the second time? Failure cases may be needed.
>
> To mitigate error accumulation, we always retain the original start and end frames as appearance references in each iteration, which prevents devastatingly diverging directions. The notion of "wrong" directions is somewhat subjective due to the multiple feasible directions. Our goal is to predict one plausible trajectory with all interpolated objects remaining clear. Exploring methods to recover any potential direction, or perhaps to learn a distribution of directions, is a direction for our future research.
>
>
>
> > Is manipulable interpolation of optical flow based VFI methods also possible?
>
> We assume that Reviewer 1WoD is referring to the traditional method of directly warping images based on optical flow. While manipulating certain regions by scaling optical flow through time interpolation is feasible, this approach is limited to strictly linear warping or motion. In addition, this approach might result in the interpolated image having unfilled gaps or “holes”.

---

> ### Author Response · Authors · 2023-11-18
>
> > Would changing to a newer optical flow estimation method result in more improvements? Especially for the exception AMT-S.
>
> Thank you for raising this interesting question!
>
> To begin, we kindly remind that AMT-S's exception occurs when using only the iterative reference-based estimation. The distance indexing can stabilize the benefits gained from iterative reference-based estimation.
>
> We utilize GMFlow [1] to precompute distance maps for comparison. The results are as follows:
>
> | Vimeo90K (GMFlow) | PSNR    | SSIM   | LPIPS  | NIQE   |
> | ----------------- | ------- | ------ | ------ | ------ |
> | RIFE [T]          | 28.2175 | 0.9116 | 0.1048 | 6.6626 |
> | RIFE [D]$_u$      | 27.2927 | 0.8983 | 0.1009 | 6.4492  |
> | RIFE [D,R]$_u$     | 26.9583 | 0.8952 | 0.0915 | 6.2799 |
>
> Our strategies still lead to consistent improvement on perceptual metrics. However, this more recent and performant optical flow estimator does not introduce improvement compared to RAFT. A likely explanation is that since we quantify the optical flow to $[0,1]$ scalar values for better generalization, our training strategies are less sensitive to the precision of the optical flow estimator.
>
>
>
> P.S., the same experimental conclusions can be derived from other SOTA models like AMT as follows:
>
> | Vimeo90k (GMFlow) | PSNR    | SSIM   | LPIPS  | NIQE   |
> | ----------------- | ------- | ------ | ------ | ------ |
> | AMT-S [T]         | 28.5202 | 0.9196 | 0.1011 | 6.8659 |
> | AMT-S [D]$_u$     | 27.0252 | 0.9004 | 0.1006 | 6.6257 |
> | AMT-S [D,R]$_u$    | 26.5586 | 0.8928 | 0.0937 | 6.4910 |
>
> We will incorporate this discussion into the final paper.
>
> Reference:
>
> [1] Xu, Haofei, Jing Zhang, Jianfei Cai, Hamid Rezatofighi, and Dacheng Tao. "Gmflow: Learning optical flow via global matching." In *Proceedings of the IEEE/CVF conference on computer vision and pattern recognition*, pp. 8121-8130. 2022.

---

> ### Comment · Reviewer_1WoD · 2023-12-05
>
> Thanks to the author's detailed responses, which I think completely addressed my concerns. I maintain my initial rating and look forward to read the updated version.

---

### Official Review · Reviewer_aknz · 2023-11-03

**Soundness:** 3 good
**Presentation:** 2 fair
**Contribution:** 2 fair
**Rating:** 6
**Confidence:** 4

**Summary:**

This work tackles video frame interpolation (VFI). In particular, it attempts to address the velocity (speed and direction) ambiguity in VFI. To address the speed ambiguity, it proposes to employ a distance map (how far the object has traveled between start and end frames), instead of optical flow, to interpolate an intermediate frame. It also proposes an iterative reference-based estimation strategy to mitigate directional ambiguity. The proposed method could be used as a plug-and-play technique and experimental results are presented on the Vimeo-90K dataset.

**Strengths:**

+ The paper attempts to address an important, yet underexplored, problem in video frame interpolation
+ The 2D manipulation of frame interpolation using segmentation models such as SAM is quite interesting
+ The paper reads fairly well
+ Supplementary video is provided

**Weaknesses:**

Major issues

* The technical formulation of the paper is flawed

    The authors propose to use a distance map (instead of optical flow) in an attempt to address the speed ambiguity in VFI. However, *instead of learning to approximate or predict the distance map*, they use the GT target frame to compute the distance map during training and use a uniform map (i.e. uniform speed) during inference. What is the point of attempting to address speed ambiguity during training, if the authors are assuming uniform speed at inference? How is the ambiguity being addressed at inference?

    The same flawed logic is used in the iterative-reference-based estimation mechanism that is proposed to address directional ambiguity. The authors use the reference frame and its distance map iteratively to interpolate intermediate frames at training time. Let's assume the authors can use the previously interpolated frame as a reference frame at inference time. However, how do the authors get the corresponding distance map? If the uniform distance map assumption is implemented at every iteration step, what is the point of doing an iterative approach? How does that address the direction ambiguity?

* Several claims in the paper are not properly motivated and convincingly justified

   The argument about Equation 4 in the main paper is not convincing. The proof in Appendix B is also based on a flawed assumption because most VFI works use L1 loss during training (not L2).

    On page 4, the authors claim, "Empirically, the model, when trained with this ambiguity, tends to produce a weighted average of possible frames during inference". This is not entirely correct. In most optical flow-based methods, the model is tasked with approximating the flow, and the intermediate frame is simply obtained by warping the input frames with the predicted flow. This also applies to kernel-based methods. Hence, the ambiguity that happens in the estimated motion is not equivalent to predicting an average of possible frames. Moreover, the claim that the ambiguity gets worse for the multi-frame interpolation scheme is also not convincing. This is because each intermediate frame is predicted by time-interpolating the estimated flow between the input frames, i.e. each intermediate frame does not have a multitude of possibilities.

    The whole point of the distance map and its benefit in addressing the speed ambiguity is also not clear. The key aspect of Equation 7 is actually the computed optical flows. why does the ratio of the flows provide extra comprehension during the training phase as the authors claim? What is the difference if we train the interpolation network by simply incorporating  $V_{0\rightarrow 1}$ and $V_{0\rightarrow t}$?

    The argument about the convergence limits in Fig 6 is also not convincing. It is likely happening not because of what is claimed, i.e. addressing velocity ambiguity. It is simply because GT information is used during training in the proposed methods while traditional training does not use GT information.

* The experimental settings are not clear and the presented results are quite limited

   Were all baseline trained using the same experimental setting? For instance, was RAFT used in all methods?

   Why did the authors only perform experimental comparisons on the Vimeo-90 K dataset? There are several VFI benchmarks that are commonly used to compare VFI methods. Why did not the authors use those benchmarks as well?

* More experimental analyses are needed to verify the benefit of the proposed method

   As the proposed distance map computation is heavily dependent on optical flow, did the authors experiment with using optical flow estimation methods other than RAFT?

   The problem the paper is trying to solve is more relevant in VFI situations where the temporal distance between input frames is relatively large. However, the authors do not show any experimental analysis in these scenarios.


Minor issues

* The technical novelty of the work is quite limited.
* Please cite and discuss some relevant works [1,2].
* What would be the added inference time by plugging the proposed method into existing methods?
* The authors repeatedly use the term, "Our observation ...", yet fail to provide convincing empirical evidence


References

[1] Exploring Motion Ambiguity and Alignment for High-Quality Video Frame Interpolation, CVPR 2023

[2] Deep Iterative Frame Interpolation for Full-frame Video Stabilization, SIGGRAPH Asia 2019

-----------------------------------------------------------------
POST REBUTTAL

I thank the authors for the well-written rebuttal. I have slightly misunderstood the main message of the work in my initial review and the authors did a great job in correcting my misunderstanding. Most of my concerns are sufficiently addressed in the rebuttal. Hence, I will happily increase my score to accept.

**Questions:**

Please refer to the questions raised in the "Weaknesses" section and try to address them carefully.

---

> ### Author Response · Authors · 2023-11-12
>
> Thanks for raising your concerns about the technical formulation, claim justification, experimental settings and analysis. Before going into the experiments, we would like to clarify some potential misunderstanding of our approach.
>
> > However, *instead of learning to approximate or predict the distance map*, they use the GT target frame to compute the distance map during training and use a uniform map (i.e. uniform speed) during inference. What is the point of attempting to address speed ambiguity during training, if the authors are assuming uniform speed at inference? How is the ambiguity being addressed at inference?
>
> We would like to provide a more intuitive and detailed explanation on the ambiguity issue in video frame interpolation (VFI) and how we resolve it, which will be integrated into the final paper.
>
> First and foremost, it's essential to clarify that **velocity ambiguity can solely exist and be resolved in the TRAINING phase, not in the INFERENCE phase**. The key idea behind our approach can be summarized as follows: While conventional VFI methods with time indexing rely on a one-to-many mapping, our distance indexing learns an approximate one-to-one mapping, which resolves the ambiguity during training. When the input-output relationship is one-to-many during training, the training process fluctuates among conflicting objectives, ultimately preventing convergence towards any specific optimization goal. In VFI, the evidence is the generation of blurry images in the inference phase. Once the ambiguity has been resolved using the new indexing method in the training phase, the model can produce significantly clearer results regardless of the inference strategy used.
>
> Indeed, this one-to-many ambiguity in training is not unique to VFI, but for a wide range of machine learning problems. In some areas, researchers have come up with similar methods.
>
>
>
> **A specific instantiation of this problem:** Let us look at an example in text-to-speech (TTS). The same text can be paired with a variety of speeches, and direct training without addressing ambiguities can result in a **"blurred" voice** (a statistical average voice). To mitigate this, a common approach is to incorporate a speaker embedding vector or a style embedding vector (representing different gender, accents, speaking styles, etc.) during training, which helps reduce ambiguity. **During the inference phase, utilizing an average user embedding vector can yield high-quality speech output.** Furthermore, by manipulating the speaker embedding vector, effects such as altering the accent and pitch can also be achieved.
> Here is a snippet from a high-impact paper which came up with the style embedding in TTS: ***“Many TTS models, including recent end-to-end systems, only learn an averaged prosodic distribution over their input data, generating less expressive speech – especially for long-form phrases. Furthermore, they often lack the ability to control the expression with which speech is synthesized.”*** from Wang, Yuxuan, et al. "Style tokens: Unsupervised style modeling, control and transfer in end-to-end speech synthesis." International conference on machine learning. PMLR, 2018, cited by 816
>
> Understanding this example can significantly help understand our paper, as there are many similarities between the two (*e.g.,* motivation, solution, and manipulation).
>
> (to be continued ...)

---

> ### Author Response · Authors · 2023-11-12
>
> **A minimal symbolic example to help readers understand better:** Assuming we want to train a mapping function $\mathcal{F}$ from numbers to characters.
>
> **Training input-output pairs with ambiguity** ($\mathcal{F}$ is optimized):
>
> $1 \stackrel{\mathcal{F}}{\longrightarrow} a$, $1 \stackrel{\mathcal{F}}{\longrightarrow} b$, $2 \stackrel{\mathcal{F}}{\longrightarrow} a$, $2 \stackrel{\mathcal{F}}{\longrightarrow} b$
>
> $\mathcal{\mathcal{F}}$ is optimized with some losses involving the input-output pairs above.
>
> $\underset{\mathcal{F}}{\min} \quad L(\mathcal{F}(1), a) + L(\mathcal{F}(1),b) + L(\mathcal{F}(2),a) +L(\mathcal{F}(2),b),$
>
> where $L$ can be L1, L2 or any other kind of losses (details on the losses are explained later).
> Because the same input is paired with multiple different outputs, the model $\mathcal{F}$ is optimized to learn an average (or, generally, a mixture) of the conflicting outputs, which results in blur at inference:
>
> Inference phase ($\mathcal{F}$ is fixed):
>
> $1 \stackrel{\mathcal{F}}{\longrightarrow} \{a,b\}?$, $2 \stackrel{\mathcal{F}}{\longrightarrow} \{a,b\}?$
>
> **Training without ambiguity** ($\mathcal{F}$ is optimized):
>
> $1 \stackrel{\mathcal{F}}{\longrightarrow} a$, $1 \stackrel{\mathcal{F}}{\longrightarrow} a$, $2 \stackrel{\mathcal{F}}{\longrightarrow} b$, $2 \stackrel{\mathcal{F}}{\longrightarrow} b$
>
> In the input-output pairs above, each input value is paired with exactly one output value. Therefore, $\mathcal{F}$ is trained to learn a unique and deterministic mapping.
>
> Inference phase ($\mathcal{F}$ is fixed):
>
> $1 \stackrel{\mathcal{F}}{\longrightarrow} a$, $2 \stackrel{\mathcal{F}}{\longrightarrow} b$
>
>
>
> **Coming back to VFI:**  When time indexing is used, the same $t$ value is paired to images where the objects are located at various locations due to the speed and directional ambiguities. When distance mapping is used, a single $d$ value is paired to images where the objects are always at the same distance ratio, which allows the model to learn a more deterministic mapping for resolving the speed ambiguity.
>
> It is important to note that fixing the ambiguity does not solve all the problems: At inference time, the "correct" (close to ground-truth) distance map is not available. In this work, we show that it is possible to provide uniform distance maps as inputs to generate a clear output video, which is not perfectly pixelwise aligned with the ground truth. This is the reason why the proposed method does not achieve state-of-the-art in terms of PSNR and SSIM (Table 1).  However, it achieves sharper frames with higher perceptual quality, which is shown by the better LPIPS and NIQE.
>
> We claim the "correct" distance map is hard to estimate accurately from merely two frames since there are a wide range of possible velocities. If considering more neighboring frames (more observation information), it is possible to estimate an accurate distance map for pixelwise aligned interpolation, which we leave for future work.
>
> Furthermore, manipulating distance maps corresponds to sampling other possible unseen velocities, *i.e.*, 2D manipulation of frame interpolation, similar to that mentioned TTS paper (Wang, Yuxuan, et al.).

---

> > ### Author Response · Authors · 2023-11-12
> >
> > > Let's assume the authors can use the previously interpolated frame as a reference frame at inference time. However, how do the authors get the corresponding distance map? If the uniform distance map assumption is implemented at every iteration step, what is the point of doing an iterative approach? How does that address the direction ambiguity?
> >
> > A uniform distance map is used at every step, which, as discussed above, is sufficient to resolve the speed ambiguity (though not pixelwise aligned).
> >
> > This divide-and-conquer idea further mitigates the directional ambiguity since it breaks a long trajectory into a series of small ones. Fig 2(b) and 3(b) gives an example: For a long trajectory, the same $t$ (or $d$) may be paired with training data where the ball appears all the way from top to bottom in the image, causing ambiguity. In the reference-based estimation, the same $d$ is paired with a much narrower range of possible locations.
> > As a result, the model learns to predict a sharper image, albeit not pixelwise aligned trajectory.
> > On a side note, the original start and end frames remain as inputs (appearance reference) to each iteration to prevent the accumulation of errors.
> >
> >
> >
> > > Equation 4 in the main paper is not convincing. The proof in Appendix B is also based on a flawed assumption because most VFI works use L1 loss during training (not L2).
> >
> > The proof was merely meant to show that the Equation 4 can be rigorously proven when L2 loss is used as a special case. For other losses, this specific equation no longer holds, but we empirically observe that the model still learns an aggregated mixture of training frames which results in blur (RIFE and EMA-VFI: Laplacian loss - L1 loss between the Laplacian pyramids of image pairs; IFRNet and AMT: Charbonnier loss). We will clarify this in the paper.
> >
> >
> >
> > >  In most optical flow-based methods, the model is tasked with approximating the flow, and the intermediate frame is simply obtained by warping the input frames with the predicted flow. This also applies to kernel-based methods. Hence, the ambiguity that happens in the estimated motion is not equivalent to predicting an average of possible frames
> >
> > It is true that the warping step is deterministic. However, ambiguity in the estimated flow itself can lead to blur. Specifically, one pixel in the input maybe forward-warped to multiple pixels in the output, resulting in blur.
> >
> >
> >
> > > Moreover, the claim that the ambiguity gets worse for the multi-frame interpolation scheme is also not convincing. This is because each intermediate frame is predicted by time-interpolating the estimated flow between the input frames, i.e. each intermediate frame does not have a multitude of possibilities.
> >
> > We would like to highlight that, to learn non-linear warping/motion, modern arbitrary-time VFI methods (*e.g.*, RIFE, EMA-VFI, IFRNet, and AMT) does **NOT** time-interpolate the estimated flow. Instead, they directly predict the flow at specified $t$. Each predicted flow map suffers from the ambiguity mentioned above.
> >
> >
> >
> > > The key aspect of Equation 7 is actually the computed optical flows. why does the ratio of the flows provide extra comprehension during the training phase as the authors claim? What is the difference if we train the interpolation network by simply incorporating $V_{0\to 1}$and $V_{0\to t}$?
> >
> > The main reason for using the ratio of the flows is that it provides a normalized value between $[0, 1]$, which gives a better-defined input as compared to the unbounded flow values. Besides this, it also makes it possible to provide a uniform map at inference time rather than running a separate optical flow estimator. We will clarify this in the paper.
> >
> >
> >
> > If there is still any confusion about our approach, we are more than welcome to discuss and offer explanations!

---

> ### Author Response · Authors · 2023-11-18
>
> Thank you for your suggestions about experiments!
>
> > Were all baseline trained using the same experimental setting? For instance, was RAFT used in all methods?
>
> Yes, the distance maps used by each model were precomputed offline using RAFT.
>
>
>
> >  Why did the authors only perform experimental comparisons on the Vimeo-90 K dataset? There are several VFI benchmarks that are commonly used to compare VFI methods. Why did not the authors use those benchmarks as well?
>
> The septuplet set of Vimeo-90K is large enough to train a practical video frame interpolation model, and it represents the situations where the temporal distance between input frames is large (Please refer to the supplementary video for a better perception). Notably, state-of-the-art methods such as AMT (CVPR'2023) have struggled to train a clear arbitrary time video frame interpolator on this extremely challenging dataset, as claimed on their GitHub:
>
> `Zhen Li et al.: We previously attempted to train on Septuplet but did not achieve good results.`
>
> Thus, Vimeo90K (septuplet) can best demonstrate the velocity ambiguity problem that our work wants to highlight. With our disambiguation strategies, state-of-the-art models can be successfully trained on this dataset.
>
> Nonetheless, we report more results on other benchmarks in the following. The following table shows the results of RIFE on Adobe240 [1] and X4K1000FPS [2] for 8x interpolation, using uniform maps.
>
>
> | Adobe240 (x8) | PSNR    | SSIM   | LPIPS  | NIQE   |
> | ------------------ | ------- | ------ | ------ | ------ |
> | RIFE [T]           | 30.2430 | 0.9386 | 0.0729 | 5.2060 |
> | RIFE [D]$_{u}$     | 30.4704 | 0.9384 | 0.0565 | 4.9735 |
> | RIFE [D,R]$_{u}$    | 30.2961 | 0.9372 | 0.0542 | 4.9068 |
>
> | X4K1000FPS (x8) | PSNR    | SSIM   | LPIPS  | NIQE   |
> | -------------------- | ------- | ------ | ------ | ------ |
> | RIFE [T]             | 36.3557 | 0.9667 | 0.0403 | 7.1302 |
> | RIFE [D]$_{u}$       | 36.7964 | 0.9642 | 0.0315 | 6.9364 |
> | RIFE [D,R]$_{u}$      | 36.2588 | 0.9642 | 0.0315 | 6.9242 |
>
> Distance indexing [D] and iterative reference-based estimation [R] strategies can consistently help improve the perceptual quality. In addition, it is noteworthy that [D]$_u$ is better than [T]$_u$ in terms of the pixel-centric metrics like PSNR, showing that the constant speed assumption (uniform distance maps) holds well on these two easier benchmarks.
>
>
> > The problem the paper is trying to solve is more relevant in VFI situations where the temporal distance between input frames is relatively large. However, the authors do not show any experimental analysis in these scenarios.
>
> As explained above, Vimeo90K (septuplet) dataset is the preferable dataset in terms of large temporal distance. We further show 16x interpolation on X4K1000FPS with larger temporal distance as follows:
>
> | X4K1000FPS (x16) | PSNR    | SSIM   | LPIPS  | NIQE   |
> | ---------------- | ------- | ------ | ------ | ------ |
> | RIFE [T]         | 31.0401 | 0.9102 | 0.1043 | 7.2152 |
> | RIFE [D]$_u$         | 31.6000 | 0.9141 | 0.0944 | 6.9531 |
> | RIFE [D,R]$_u$        | 31.5195 | 0.9220 | 0.0786 | 6.9265 |
>
> The results highlight that the benefits of our strategies are more pronounced with increased temporal distances.
>
> > As the proposed distance map computation is heavily dependent on optical flow, did the authors experiment with using optical flow estimation methods other than RAFT?
>
> We use GMFlow [3] to precompute distance maps for comparison. Our strategies still yield consistent improvement on perceptual metrics, as detailed below:
>
> | Vimeo90K (GMFlow) | PSNR    | SSIM   | LPIPS  | NIQE   |
> | ----------------- | ------- | ------ | ------ | ------ |
> | RIFE [T]          | 28.2175 | 0.9116 | 0.1048 | 6.6626 |
> | RIFE [D]$_u$      | 27.2927 | 0.8983 | 0.1009 | 6.4492 |
> | RIFE [D,R]$_u$     | 26.9583 | 0.8952 | 0.0915 | 6.2799 |
>
> We will incorporate all the above content into the final paper.
>
> (to be continued ...)

---

> ### Author Response · Authors · 2023-11-18
>
> P.S., the same experimental conclusions can be derived from other state-of-the-art models like AMT as follows:
>
> | Adobe240 (x8) | PSNR    | SSIM   | LPIPS  | NIQE   |
> | ------------- | ------- | ------ | ------ | ------ |
> | AMT [T]       | 29.5985 | 0.9333 | 0.0935 | 5.4004 |
> | AMT [D]$_u$       | 29.7449 | 0.9341 | 0.0700 | 5.0037 |
> | AMT [D,R]$_u$      | 29.4671 | 0.9329 | 0.0630 | 4.9392 |
>
> | X4K1000FPS (x8) | PSNR    | SSIM   | LPIPS  | NIQE   |
> | --------------- | ------- | ------ | ------ | ------ |
> | AMT [T]         | 35.6691 | 0.9612 | 0.0598 | 7.3296 |
> | AMT [D]$_u$         | 36.6280 | 0.9631 | 0.0412 | 7.0355 |
> | AMT [D,R]$_u$        | 36.3044 | 0.9645 | 0.0355 | 6.8704 |
>
> | X4K1000FPS (x16) | PSNR    | SSIM   | LPIPS  | NIQE   |
> | ---------------- | ------- | ------ | ------ | ------ |
> | AMT [T]          | 31.6443 | 0.9242 | 0.1222 | 7.6546 |
> | AMT [D]$_u$      | 32.5098 | 0.9352 | 0.0816 | 7.1798 |
> | AMT [D,R]$_u$       | 31.9543 | 0.9311 | 0.0771 | 7.0707 |
>
> | Vimeo90k (GMFlow) | PSNR    | SSIM   | LPIPS  | NIQE   |
> | ----------------- | ------- | ------ | ------ | ------ |
> | AMT-S [T]         | 28.5202 | 0.9196 | 0.1011 | 6.8659 |
> | AMT-S [D]$_u$     | 27.0252 | 0.9004 | 0.1006 | 6.6257 |
> | AMT-S [D,R]$_u$    | 26.5586 | 0.8928 | 0.0937 | 6.4910 |
>
>
>
> > The technical novelty of the work is quite limited.
>
> > The authors repeatedly use the term, "Our observation ...", yet fail to provide convincing empirical evidence
>
> It may appear at first glance that this work lacks technical novelty since we do not propose any novel network architecture. However, we claim this is indeed the strength of our approach: By replacing time indexing with distance indexing as network inputs, the proposed approach can be applied to a wide range of VFI methods, driving consistent improvement in perceptual quality. We hope that through our additional explanation about our approach, Reviewer aknz can recognize our motivation and novelty as other reviewers did (Reviewer 1WoD, PtYf, and BPE4).
>
>
>
> > Please cite and discuss some relevant works [1,2].
> >
> > [1] Exploring Motion Ambiguity and Alignment for High-Quality Video Frame Interpolation, CVPR 2023
> >
> > [2] Deep Iterative Frame Interpolation for Full-frame Video Stabilization, SIGGRAPH Asia 2019
>
> We appreciate the reminder and would like to point out that our paper does include a discussion about [1] in the second paragraph of the "Learning Paradigms" section. Additionally, we will incorporate a discussion of [2] in our final paper, which highlights the use of VFI techniques in an iterative manner for video stabilization.
>
>
>
> > What would be the added inference time by plugging the proposed method into existing methods?
>
> **Distance indexing**: Transitioning from time indexing ($[T]$) to distance indexing ($[D]$) does not introduce extra computational costs during the inference phase, yet significantly enhancing image quality. So, the inference time is the same.
>
> **Iterative reference-based estimation ($[D,R]$)**: Given that the computational overhead of merely expanding the input channel, while keeping the rest of the structure unchanged, is negligible, the total inference time is equal to the number of iterations $\times$ the consumption of the $[D]$ model. We would like to highlight that this iterative strategy is optional: Users can adopt this strategy at will when optimal interpolation results are demanded and the computational budget allows. Our experiments (Table 2) show that 2 iterations, *i.e.,* doubling the computational cost, yield cost-effective improvements. We will include these details in our final paper for clarity.
>
>
>
> References:
>
> [1] Su, Shuochen, Mauricio Delbracio, Jue Wang, Guillermo Sapiro, Wolfgang Heidrich, and Oliver Wang. "Deep video deblurring for hand-held cameras." In *Proceedings of the IEEE conference on computer vision and pattern recognition*, pp. 1279-1288. 2017.
>
> [2] Sim, Hyeonjun, Jihyong Oh, and Munchurl Kim. "Xvfi: extreme video frame interpolation." In *Proceedings of the IEEE/CVF international conference on computer vision*, pp. 14489-14498. 2021.
>
> [3] Xu, Haofei, Jing Zhang, Jianfei Cai, Hamid Rezatofighi, and Dacheng Tao. "Gmflow: Learning optical flow via global matching." In *Proceedings of the IEEE/CVF conference on computer vision and pattern recognition*, pp. 8121-8130. 2022.

---

### Public Comment · ~Zhewei_Huang1 · 2023-11-13

Resolving ambiguities in frame interpolation learning and an editable frame interpolation method sounds very impressive.

I have researched related issues, and the author's research is much deeper than mine. Unfortunately, although I strongly bid on this paper, I am not a reviewer.

I have some small suggestions:
In Appendix B, it seems more reasonable to use L1 loss?

If the author focuses on visual effects, it would be meaningful to properly combine and compare the methods proposed in this paper with vgg loss.

---

> ### Author Response · Authors · 2023-11-22
>
> Thank you, Zhewei, for your generous recognition of our work and for providing an excellent baseline (RIFE) for our research.
>
> > In Appendix B, it seems more reasonable to use L1 loss?
>
> Thank you for your suggestion. The proof was merely meant to show that the Equation 4 can be rigorously proven when L2 loss is used as a special case. For other losses, this specific equation no longer holds, but we empirically observe that the model still learns an aggregated mixture of training frames which results in blur (RIFE and EMA-VFI: Laplacian loss - L1 loss between the Laplacian pyramids of image pairs; IFRNet and AMT: Charbonnier loss). We will clarify this in the paper.
>
> > If the author focuses on visual effects, it would be meaningful to properly combine and compare the methods proposed in this paper with vgg loss.
>
> We present the results of employing the more recent LPIPS loss [1] with a VGG backbone [2], as shown in the table below. Since [T] is directly trained to optimize the LPIPS loss at timestep $t$, it achieves slightly better LPIPS than [D]$_u$ and [D,R]$_u$, which use the uniform distance map as an approximation. Nevertheless, [D]$_u$ and [D,R]$_u$ yield significant improvement on the non-reference perceptual quality metric NIQE, consistently demonstrating the effectiveness of our strategies in resolving velocity ambiguity.
>
> | Vimeo90k (VGG-based LPIPS loss) | PSNR    | SSIM   | LPIPS  | NIQE   |
> | ------------------------------- | ------- | ------ | ------ | ------ |
> | RIFE [T]                        | 27.1860 | 0.8976 | 0.0609 | 6.3071 |
> | RIFE [D]$_u$                    | 26.7137 | 0.8889 | 0.0649 | 5.9012 |
> | RIFE [D,R]$_u$                   | 26.7170 | 0.8902 | 0.0644 | 5.8368 |
>
>
>
> [1] Zhang, Richard, Phillip Isola, Alexei A. Efros, Eli Shechtman, and Oliver Wang. "The unreasonable effectiveness of deep features as a perceptual metric." In *Proceedings of the IEEE conference on computer vision and pattern recognition*, pp. 586-595. 2018.
>
> [2] Simonyan, Karen, and Andrew Zisserman. "Very deep convolutional networks for large-scale image recognition." *arXiv preprint arXiv:1409.1556* (2014).

---

### Author Response · Authors · 2023-11-20

We further include source code and detailed instructions in the supplementary material to support the reproducibility of our work.